# QuantProb: Generalizing Probabilities along with Predictions for a Pre-trained Classifier

**Aditya Challa**[1]     **Snehanshu Saha**[1]     **Soma S. Dhavala**[2]

[1]Department of CSIS and APPCAIR , Birla Institute of Technology and Science, Goa, India ,
[2]Director-ML, Wadhwani AI, Bengaluru, India ,

## Abstract

Quantification of Uncertainty in predictions is a challenging problem. In the classification settings, although deep learning based models generalize well, class probabilities often lack reliability. Calibration errors are used to quantify uncertainty, and several methods exist to minimize calibration error. We argue that between the choice of having a minimum calibration error on original distribution which increases across distortions or having a (possibly slightly higher) calibration error which is constant across distortions, we prefer the latter We hypothesize that the reason for unreliability of deep networks is - The way neural networks are currently trained, the probabilities do not generalize across small distortions. We observe that quantile based approaches can potentially solve this problem. We propose an innovative approach to decouple the construction of quantile representations from the loss function allowing us to compute quantile based probabilities without disturbing the original network. We achieve this by establishing a novel duality property between quantiles and probabilities, and an ability to obtain quantile probabilities from any pre-trained classifier.

While post-hoc calibration techniques successfully minimize calibration errors, they do not preserve robustness to distortions. We show that, Quantile probabilities (QuantProb), obtained from Quantile representations, preserve the calibration errors across distortions, since quantile probabilities generalize better than the naive Softmax probabilities.

## 1   INTRODUCTION

Deep learning models have become ubiquitous across diverse domains, and are increasingly being used for several critical applications. However, in practice, when dealing with ML systems, it is important that we capture the uncertainty in the prediction along with the predictions themselves. As noted in Guo et al. [2017], deep networks tends to be overconfident in their predictions. Well behaved probabilities can also help in answering common questions which arise in practice - (a) Can this model be used on the given data input? and (b) If so, how much can one trust the probability prediction obtained? The former refers to the problem of Out-of-Distribution (OOD) detection [Hendrycks and Gimpel, 2017, Fort et al., 2021] and the latter refers to the problem of Calibration [Guo et al., 2017, Lakshminarayanan et al., 2017, Liu et al., 2020]. Understanding the applicability of a given deep learning model is a topic of current research [Ribeiro et al., 2016, Farrell et al., 2021, Nguyen et al., 2015, Jiang et al., 2018].

As Kumar et al. [2022] argues, calibration of models can also help in improving OOD accuracy. In this article we consider the quantile regression based approach to provide better estimates of the uncertainty.

**Quantile regression** techniques [Koenker, 2005, Kordas, 2006] provide much richer information about the model, allowing for more comprehensive analysis and understanding relationship between different variables. In Tagasovska and Lopez-Paz [2019], the authors show how simultaneous quantile regression (SQR) techniques can be used to estimate the uncertainties of the deep learning model in the case of regression problems. However, these techniques aren't widely adopted in modern deep learning based systems since the loss function is restricted to be mean absolute error (MAE) or the pinball loss which is difficult to optimize in the case of classification problem. Moreover, MAE loss might not compatible with domain specific losses [Chung et al., 2021].

**Problem Statement:**   Consider the problem setting where a pre-trained classifier $f_\theta(\boldsymbol{x})$ (including the dataset on which it is trained) is given and we wish to assign meaningful probabilities to the prediction. The naive approach is to use softmax outputs as probabilities. However, softmax

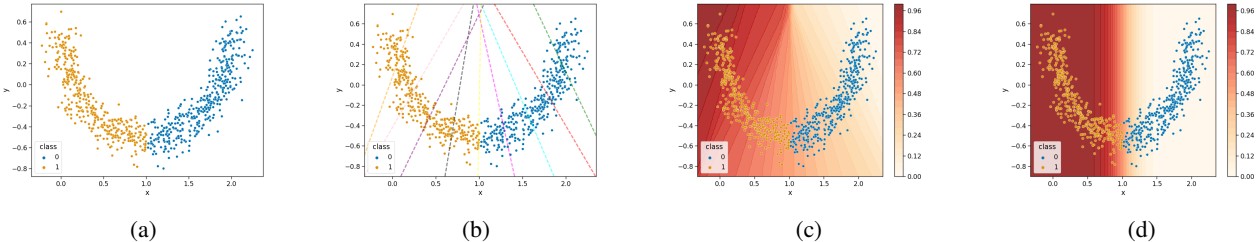

| (a) | (b) | (c) | (d) |

Figure 1: Illustrating the construction of Quantile Representations. (a) Simple toy example. (b) Illustrates different classifiers obtained for different $\tau$. (c) Quantile Probabilities Heatmap. (d) Baseline Probabilities Heatmap. Note that quantile probabilities capture the inherent structure of the dataset, while baseline probabilities only rely on distance from the boundary.

probabilities do not generalize well across small distortions. So we ask the question - *Can we assign the probabilities such that calibration error remains constant (possibly not zero) across distortions?* To our knowledge, there exists no method which can achieve this. *This is one of the open questions posed in Kumar et al. [2019].*

**(Motivation) Minimizing Calibration Error vs Making Calibration Errors robust to distortions:** It has been shown in the past that model suffer greatly due to poor calibration [Hoorde et al., 2015, van der Ploeg et al., 2016]. Some even labeled calibration error as the 'Achilles heel' of predictive analytics [Shah et al., 2018]. Reporting on calibration performance is recommended by the TRIPOD (Transparent Reporting of a multi variable prediction model for Individual Prognosis Or Diagnosis) guidelines for prediction modeling studies [Collins et al., 2015].

What does the term "well-calibrated" mean? – Ideally, one would like to have minimum calibration error across distortions. However, between the choice of having a minimum calibration error which increases across distortions or having a (slightly higher) calibration error which is constant across distortions, we prefer the latter. This is because, having constant calibration error can give us guarantees which changing error cannot. However, one should note that there is possibly a tradeoff, in the sense that if the calibration is very high but constant across distortions, then it would not be preferable.

**Overview And Contributions:** Using quantile loss function (equation 2, [Tagasovska and Lopez-Paz, 2019]) to retrain the network would hinder the purpose of assigning meaningful probabilities, since the retrained network would have different properties compared to the original. Our first contribution is to *decouple the construction of quantile representations from the loss function*. To achieve this, we establish a novel *Duality Property* between quantiles and probabilities. We then leverage the duality to construct the quantile representations for any pre-trained classifier $f_\theta$ and

consequently obtain quantile probabilities (QUANTPROB). In section 4, we show that the obtained QUANTPROB are robust to distortions, while the baseline softmax probabilities are not. Interestingly, we also show that the usual approaches to calibration such as Platt-Scaling actually make the probabilities less invariant to distortions. In the appendix we also illustrate other applications of QUANTPROB such as OOD Detection and identifying the distribution shift.

**Illustrating the Construction of QUANTPROB:** Before diving into the details, we illustrate our construction using a simple toy example. Figure 1a shows a simple toy example with 2 classes - $0, 1$. To get the quantile representation - (step 1) we first construct a simple classifier, $f_\theta(\boldsymbol{x})$ to differentiate classes $0, 1$, (step 2) To get a classifier at quantile $\tau$, construct $y_{i,\tau}^+ = I[f_\theta(\boldsymbol{x}) > \tau]^1$. Construct a classifier, $\{f_{\tau,\theta}\}$, using the new labels $y_{i,\tau}^+$. Figure 1b illustrates the classifiers obtained at different quantiles, $\tau$. (Step 3) To obtain the quantile probabilities (QUANTPROB) we use the average number of times $f_{\tau,\theta}$ predicts 1 - $\text{Avg}_\tau(I[f_{\tau,\theta} > 0.5])$. Figures 1c shows the probability heatmap obtained. Comparing it to the baseline in figure 1d, we see that QUANTPROB capture the inherent structure of the data while the baseline only considers distance from the boundary.

**Important takeaway:** One can think of QUANTPROB as obtaining level curves for the baseline probabilities. While the naive approach is to consider level curves which are parallel to the original boundary, QUANTPROB uses the data to infer the shape of these level curves, so that it reflects the shape of the underlying manifold.

## 2 SIMULTANEOUS BINARY QUANTILE REGRESSION (SBQR)

In this section, we review some of the theoretical foundations required for constructing quantile representations. For

---

[1] $I[.]$ indicates the indicator function

more details please refer to [Koenker, 2005, Kordas, 2006, Tagasovska and Lopez-Paz, 2019].

Let $p_{\text{data}}(X, Y)$, denote the distribution from which the data is generated. $X$ denotes the features and $Y$ denotes the targets (class labels). A classification algorithm predicts the latent variable (a.k.a *logits*) $Z$ which are used to make predictions on $Y$.

Let $\boldsymbol{x} \in \mathbb{R}^d$ denote the $d$ dimensional features and $y \in \{0, 1, \cdots, k\}$ denote the class labels (targets). We assume that the training set consists of $N$ i.i.d samples $\mathcal{D} = \{(\boldsymbol{x}_i, y_i)\}$. Let $\boldsymbol{z}_i = f_{\ell, \theta}(\boldsymbol{x}; \theta)$ denote the classification model which predicts the logits $\boldsymbol{z}_i$. In binary case ($k = 1$), applying the $\sigma$ (Sigmoid) function we obtain the probabilities, $p_i = f_\theta(\boldsymbol{x}_i) = \sigma(f_{\ell, \theta}(\boldsymbol{x}_i))$. For multi-class classification we use the $\text{softmax}(f_{\ell, \theta}(\boldsymbol{x}_i))$ to obtain the probabilities. The final class predictions are obtained using the $\arg\max_k p_{i,k}$, where $k$ denotes the class-index.

## 2.1 REVIEW - QUANTILE REGRESSION AND BINARY QUANTILE REGRESSION

Observe that, for binary classification, $Z$ denotes a one dimensional distribution. $F_Z(\boldsymbol{z}) = P(Z \leq \boldsymbol{z})$ denotes the cumulative distribution of a random variable $Z$. The function $F_Z^{-1}(\tau) = \inf\{\boldsymbol{z} : F_Z(\boldsymbol{z}) \geq \tau\}$ denotes the quantile distribution of the variable $Z$, where $0 < \tau < 1$. The aim of quantile regression is to predict the $\tau^{th}$ quantile of the latent variable $Z$ from the data. That is, we aim to estimate $F_Z^{-1}(\tau \mid X = \boldsymbol{x})$. Minimizing pinball-loss or check-loss [Koenker, 2005],

$$\text{pinball loss} = \sum_{i=1}^n \rho(f_\theta(\boldsymbol{x}_i), y_i; \tau)$$

$$\text{where, } \rho(\hat{y}, y; \tau) = \begin{cases} \tau(y - \hat{y}) & \text{if } (y - \hat{y}) > 0 \\ (1 - \tau)(\hat{y} - y) & \text{otherwise} \end{cases}$$

(1)

allows us to learn $f_\theta$ which estimates the $\tau^{th}$ quantile of $Y$. When $\tau = 0.5$, we obtain the loss to be equivalent to mean absolute error (MAE). For the multi-class case we follow the one-vs-rest procedure to learn quantiles for each class.

**Simultaneous Quantile Regression (SQR):** Observe that the loss in equation 1 is for a single $\tau$. Tagasovska and Lopez-Paz [2019] argues that - minimizing the expected loss over all $\tau \in (0, 1)$ where the solution depends on $\tau$,

$$\min_\psi \mathbb{E}_{\tau \sim U[0,1]}[\rho(\psi(\boldsymbol{x}, \tau), y; \tau)] \quad (2)$$

is better than optimizing for each $\tau$ separately. Using the loss in equation 2 instead of equation 1 biases the solution to have *monotonicity property*. If $\mathcal{Q}(\boldsymbol{x}, \tau)$ denotes the solution to equation 2, monotonicity requires

$$\mathcal{Q}(\boldsymbol{x}, \tau_i) \leq \mathcal{Q}(\boldsymbol{x}, \tau_j) \Leftrightarrow \tau_i \leq \tau_j \quad (3)$$

Observe that for a given $\boldsymbol{x}_i$, the function $\mathcal{Q}(\boldsymbol{x}_i, \tau)$ can be interpreted as a (continuous) representation of $\boldsymbol{x}_i$ as $\tau$ varies over $(0, 1)$. The function $\mathcal{Q}(\boldsymbol{x}, \tau)$ is referred to as *quantile representation*. $\mathcal{Q}(\boldsymbol{x}, \tau)$ is sometimes written as $\mathcal{Q}(\boldsymbol{x}, \tau; \theta)$, where $\theta$ indicates the parameters (such as weights in a neural neural network). For brevity, we do not include the parameters $\theta$ in this article unless explicitly required.

**Remark on Notation:** To differentiate between the latent scores (logits) and probabilities - we use $\mathcal{Q}(\boldsymbol{x}, \tau)$, $f_\theta(\boldsymbol{x})$ to denote the probabilities and $\mathcal{Q}_\ell(\boldsymbol{x}, \tau)$, $f_{\ell, \theta}(\boldsymbol{x})$ to denote the latent scores. Since we have the relation $\mathcal{Q}(\boldsymbol{x}, \tau) = \sigma(\mathcal{Q}_\ell(\boldsymbol{x}, \tau))$ and $f_\ell(\boldsymbol{x}) = \sigma(f_{\ell, \theta}(\boldsymbol{x}))$ and $\sigma(.)$ is monotonic, these quantities are related by a monotonic transformation.

**Why Quantile Regression?** Quantile regression techniques are relatively less adopted in the machine learning community, but offers a wide range of advantages over the traditional single point regression. Quantiles give information about the shape of the distribution, in particular if the distribution is skewed. They are robust to outliers, can model extreme events, capture uncertainty in predictions. Quantile regression techniques have been used for pediatric medicine, survival and duration time studies, discrimination and income inequality. (See supplementary material for a more thorough discussion.)

# 3 QUANTPROB: QUANTILE REPRESENTATIONS FOR PRE-TRAINED CLASSIFIER

As discussed earlier, minimizing equation 2 does not preserve the properties of the pre-trained classifier. Thus, we require a procedure to construct quantile representations without resorting to minimizing equation 2. In this section we present *duality* property of the quantile representations, which allows us to do this.

## 3.1 DUALITY BETWEEN QUANTILES AND PROBABILITIES

Observe that, for binary classification, equation 1 can be written as

$$\rho(\hat{y}, y; \tau) = \begin{cases} \tau(1 - \hat{y}) & \text{if } y = 1 \\ (1 - \tau)(\hat{y}) & \text{if } y = 0 \end{cases} \quad (4)$$

Thus the following property holds :

$$\rho(\hat{y}, y; \tau) = \rho(1 - \tau, y; 1 - \hat{y}) \quad (5)$$

We refer to the above property as *duality between quantiles and probabilities*. Let $\mathcal{Q}(\boldsymbol{x}, \tau)$ denotes a solution to equation 2. Suppose $\mathcal{Q}(\boldsymbol{x}, \tau_0) = p_i$, then $p_i$ denotes the

probability that $\boldsymbol{x}$ belongs to class 1. But from equation 5, this can also be interpreted as - $(1 - p_i)$ is the quantile at which the probability is $(1 - \tau_0)$. We exploit this interpretation to frame Algorithm 1.

More formally, we construct the empirical versions of **quantile representations** $\mathcal{Q}(\boldsymbol{x}, \tau)$, which given the quantile returns the probability, as

$$\mathcal{Q}(\boldsymbol{x}, \tau) = \arg\min_{\hat{y}(\boldsymbol{x})} \frac{1}{N} \sum_{i=1}^{N} \rho(\hat{y}(\boldsymbol{x}_i), \boldsymbol{y}_i, \tau) \quad (6)$$

and **probability representations** $\mathcal{P}(\boldsymbol{x}, p)$, which given a probability return the quantile, as

$$\mathcal{P}(\boldsymbol{x}, p) = \arg\min_{\hat{y}(\boldsymbol{x})} \frac{1}{N} \sum_{i=1}^{N} \rho(p, \boldsymbol{y}_i, \hat{y}(\boldsymbol{x}_i)) \quad (7)$$

**Remark:** For notational simplicity, and to make the relation explicit we use $\hat{y}(\boldsymbol{x}_i)$ instead of $\hat{y}$.

We can then derive the relation between the quantile and probability representations as follows - Say we have that $\mathcal{Q}(\boldsymbol{x}, \tau^*) = p_k$, for some $\boldsymbol{x}$

$$\begin{aligned}
\mathcal{Q}(\boldsymbol{x}, 1 - p_k) &= \arg\min_{\hat{y}(\boldsymbol{x})} \frac{1}{N} \sum_{i=1}^{N} \rho(\hat{y}(\boldsymbol{x}_i), \boldsymbol{y}_i, 1 - p_k) \\
&= \arg\min_{\hat{y}(\boldsymbol{x})} \frac{1}{N} \sum_{i=1}^{N} \rho(p_k, \boldsymbol{y}_i, 1 - \hat{y}(\boldsymbol{x}_i)) \\
&= 1 - \arg\min_{\hat{y}(\boldsymbol{x})} \frac{1}{N} \sum_{i=1}^{N} \rho(p_k, \boldsymbol{y}_i, \hat{y}(\boldsymbol{x}_i)) \\
&= 1 - \mathcal{P}(\boldsymbol{x}, p_k)
\end{aligned} \quad (8)$$

The interesting thing to note about the above equation is that, the LHS - $\mathcal{Q}(\boldsymbol{x}_i, 1 - p_k)$ denotes the probability at quantile $1 - p_k$, while the RHS - $1 - \mathcal{P}(\boldsymbol{x}_i, p_k)$ denotes the quantile at probability $p_k$. It is easy to see that the monotonicity property of quantiles in equation 3, extends to the monotonicity property of probability representations,

$$p_1 \leq p_2 \Leftrightarrow \mathcal{P}(x_i, p_1) \leq \mathcal{P}(x_i, p_2) \quad (9)$$

**Strong Duality:** To illustrate the power of this observation, if we have a strong one-one relationship between the quantiles and probabilities, that is, for each $\boldsymbol{x}$, the function $Q(\boldsymbol{x}, .)$ is bijective and also satisfies,

$$\begin{aligned}
\mathcal{Q}(\boldsymbol{x}, \tau) &= \arg\min_{\hat{y}(\boldsymbol{x})} \frac{1}{N} \sum_{i=1}^{N} \rho(\hat{y}(\boldsymbol{x}_i), \boldsymbol{y}_i, \tau) \\
\mathcal{Q}^{-1}(\boldsymbol{x}, p) &= \arg\min_{\hat{y}(\boldsymbol{x})} \frac{1}{N} \sum_{i=1}^{N} \rho(p, \boldsymbol{y}_i, \hat{y}(\boldsymbol{x}_i))
\end{aligned} \quad (10)$$

Then, in this special case we have $\mathcal{Q}(\boldsymbol{x}_k, \tau^*) = p_k \Leftrightarrow \mathcal{Q}(\boldsymbol{x}_k, 1 - p_k) = 1 - \mathcal{Q}^{-1}(\boldsymbol{x}_k, p_k) = 1 - \tau^*$. We refer to this as *Strong Duality*.

---

**Algorithm 1** Generating Quantile Representations.

- Let $\mathcal{D} = \{(\boldsymbol{x}_i, y_i)\}$ denote the training dataset. Assume that a pre-trained binary classifier $f_\theta(\boldsymbol{x})$ is given. The aim is to generate the quantile representations with respect to $f_\theta(\boldsymbol{x})$. We refer to this $f_\theta(\boldsymbol{x})$ as base-classifier.

- Define $y_{i,\tau}^+ = I[f_\theta(\boldsymbol{x}_i) > (1 - \tau)]$. We refer to this as modified labels at quantile $\tau$.

- To obtain $\mathcal{Q}(\boldsymbol{x}, \tau)$, train the classifier using the dataset $\mathcal{D}_\tau^+ = \{((\boldsymbol{x}_i, \tau), y_{i,\tau}^+)\}$, for all values of $\tau$ simultaneously. That is, for the input $(\boldsymbol{x}_i, \tau)$ the classifier is trained to predict $y_{i,\tau}^+$.

---

The main implication being – *If we have information about the median solution, $\mathcal{Q}(\boldsymbol{x}, 0.5)$, and sufficient data, then we can obtain $\mathcal{Q}(\boldsymbol{x}, \tau)$ by constructing a classifier with labels, $y(\boldsymbol{x}) = 1 \Leftrightarrow \mathcal{Q}(\boldsymbol{x}, 0.5) \geq 1 - \tau$.*

**Why does algorithm 1 return quantile representations?** Assume for an arbitrary $\boldsymbol{x}_i$, we have $\mathcal{Q}(\boldsymbol{x}_i, 0.5) = p_i$. Then, thanks to duality we have, $\mathcal{P}(\boldsymbol{x}_i, 0.5) = 1 - p_i$. Then, monotonicity in equation 9 implies – if we have if the probability is less than 0.5, then the corresponding quantile $\tau \leq 1 - p_i$ and if probability is greater than 0.5, we have that the corresponding quantile $\tau \geq 1 - p_i$.

In other words, at a given quantile $\tau$, $\boldsymbol{x}_i$ will belong to class 1 if $\tau > (1 - p_i) \Leftrightarrow p_i > (1 - \tau) \Leftrightarrow f_\theta(\boldsymbol{x}_i) > (1 - \tau)$. Defining, $y_{i,\tau}^+ = I[f_\theta(\boldsymbol{x}_i) > (1 - \tau)]$, we have that the classifier at quantile $\tau$ fits the data $\mathcal{D}_\tau^+ = \{((\boldsymbol{x}_i, \tau), y_{i,\tau}^+)\}$ and thus can be used to identify $\mathcal{Q}(\boldsymbol{x}, \tau)$. This gives us the algorithm 1 to get the quantile representations for an arbitrary classifier $f_\theta(\boldsymbol{x})$.

Specifically, we have the following theorem

**Theorem 3.1** *Let $\psi^*$ denote a minimizer of the following cost,*

$$\arg\min_{\psi} \mathbb{E}_{\tau \in U[0,1]} \left[ \frac{1}{N} \sum_{i=1}^{N} \rho(I[\psi(\boldsymbol{x}_i, \tau) \geq 0.5], y_i; \tau) \right] \quad (11)$$

*over the dataset $\mathcal{D}$. Then, the solution $\mathcal{Q}(\boldsymbol{x}, \tau)$ obtained by algorithm 1 with the base classifier as $\psi^*(\boldsymbol{x}, 0.5)$, minimizes the cost in equation 11 as well, assuming strong duality for $\mathcal{Q}(\boldsymbol{x}, \tau)$.*

**Remark**: We assume that the hypothesis class of $f_\theta$, $\mathcal{Q}(\boldsymbol{x}, \tau)$ are large to enough to fit any finite datasets. For instance we can consider these to be large over-parameterized neural networks. Note that, in comparison with equation 2, equation 11 has an additional indicator function on top of the sigmoid function. So, algorithm 1 gives a solution only upto this approximation. The proof for the above theorem is discussed in the supplementary material.

**Duality - Importance and Intuition:** Algorithm 1 and theorem 3.1 hinges on the duality property. Recall that pinball loss equation 4 penalizes the positive errors and negative errors differently. In the case of binary classification, since $f_\theta(\boldsymbol{x}) \in (0,1)$, positive errors occur for class 1 and negative errors occur for class 0. Hence, the quantile value implicitly controls the probability of class 1, giving the duality property.

Thus, using quantile value as an input allows us to control the probabilities and hence confidence of our predictions. This is exploited to construct quantile representations without resorting to optimizing equation 2. This ensures that the properties of the pre-trained model are preserved while still being able to compute quantile representations.

**Remark:** The other alternate to computing quantile representations are the Bayesian approaches [Jospin et al., 2022]. It is known that computing the *full predictive distribution -* $p(y|\mathcal{D}, x) = \int p(y|w, x)p(w|\mathcal{D})dw$ is computationally difficult. Quantile representations approximate the inverse of the c.d.f of the predictive distribution for the binary classification.

To summarize, thanks to the duality in equation 5, one can compute the quantile representations for any arbitrary pre-trained classifier without modifying its behaviour. This allows for detailed analysis of the classifier and the features learned. In the following section we first discuss the implementation of algorithm 1 in practice and empirically validate the probabilities for calibration and OOD Detection.

## 3.2 GENERATING QUANTILE REPRESENTATIONS IN PRACTICE

Let $f_\theta(\boldsymbol{x})$ denote a pre-trained classifier. Given a dataset $\mathcal{D} = \{(\boldsymbol{x}_i, y_i)\}_i$, we construct a *quantile dataset -* $\{((\boldsymbol{x}_i, \tau), y^+_{i,\tau})\}_{i,\tau}$ as described in algorithm 1 with the following modifications.

**Getting $y^+_{i,\tau}$ in practice:** Instead of computing $y^+_{i,\tau} = I[f_\theta(\boldsymbol{x}) > (1 - \tau)]$, we obtain the labels using the $\tau^{th}$ quantile of logits

$$I[f_{\ell,\theta}(\boldsymbol{x}) > (1 - \tau)^{th} \text{ quantile of } \{f_{\ell,\theta}(\boldsymbol{x}_i)\}_i] \quad (12)$$

As multi-class classification problem gives class imbalance under one-vs-rest paradigm, we compute *weighted-quantiles*, where weights are assigned such that the number of samples with $f_{\ell,\theta}(\boldsymbol{x}_i) > 0$, and number of samples with $f_{\ell,\theta}(\boldsymbol{x}_i) \leq 0$ is balanced. While this assumption might lead to some bias, it allows us to circumvent the precision issues of the sigmoid function. Moreover, as we shall shortly illustrate, these probabilities are more robust compared to the naive probabilities.

**Consider only finite number of quantiles:** We only consider a fixed finite number of quantiles. The $n_\tau$ quantiles

are given by $\{1/n_\tau+1, 2/n_\tau+1, \cdots, n_\tau/n_\tau+1\}$.

For the sake of valid experimentation and comparison, we model $\mathcal{Q}(\boldsymbol{x}, \tau)$ using the same network as $f_\theta(\boldsymbol{x})$, except for the first layer. We concatenate the value of $\tau$ to the input, resulting in slightly more number of parameters in the first layer. For efficient optimization we start the training with the weights of the pre-trained classifier $f_\theta(\boldsymbol{x})$, except for the first layer. (**Remark:** However, we note that using a larger network could potentially improve the results)

**Loss function to train $\mathcal{Q}_\ell(\boldsymbol{x}, \tau)$:** Recall that $\mathcal{Q}_\ell(\boldsymbol{x}, \tau)$ indicates the latent logits. We use `BinaryCrossEntropy` loss to train $\mathcal{Q}_\ell(\boldsymbol{x}, \tau)$ where the targets are given by the modified labels $\{y^+_{i,\tau}\}$.

**Inference using $\mathcal{Q}_\ell(\boldsymbol{x}, \tau)$:** After training, we compute the probabilities as follows

$$
\begin{aligned}
p_i &= \int_{\tau=0}^{1} I[\mathcal{Q}_\ell(\boldsymbol{x}_i, \tau) \geq 0]d\tau \\
&\approx \frac{1}{n_\tau} \sum_i I[\mathcal{Q}_\ell(\boldsymbol{x}_i, \tau) \geq 0]
\end{aligned}
\quad (13)
$$

We refer to these as quantile probabilities (QUANTPROB). **Remark:** For multi-class classification, we follow a one-vs-rest approach. Hence the loss in this case would be sum of losses over all individual classes. The probability, in multi-class case, is taken to be $\arg\max_k p_{i,k}$. Note that the probabilities $p_{i,k}$ do not necessarily add up to 1 over all classes.

# 4 USING QUANTPROB FOR CALIBRATION

Recall that the key question in this article which we would like to address is - *Is there an approach to assign probabilities which can generalize better?*. To evaluate the generalizability, we consider the *calibration error* as an evaluation. If the probabilities generalize well, then one expect that the calibration error to be constant across distortions.

**Overview of Calibration:** For several applications the confidence of the predictions is important. This is measured by considering how well the output probabilities from the model reflect it's predictive uncertainty. This is referred to as *Calibration*.

Several methods [Platt, 2000, Zadrozny and Elkan, 2002, Lakshminarayanan et al., 2017, Angelopoulos et al., 2021, Liu et al., 2020] are used to improve the calibration of the deep learning models. Most of these methods consider a part of the data (apart from train data) to adjust the probability predictions. However, in [Ovadia et al., 2019, Minderer et al., 2021] it has been shown that most of the calibration

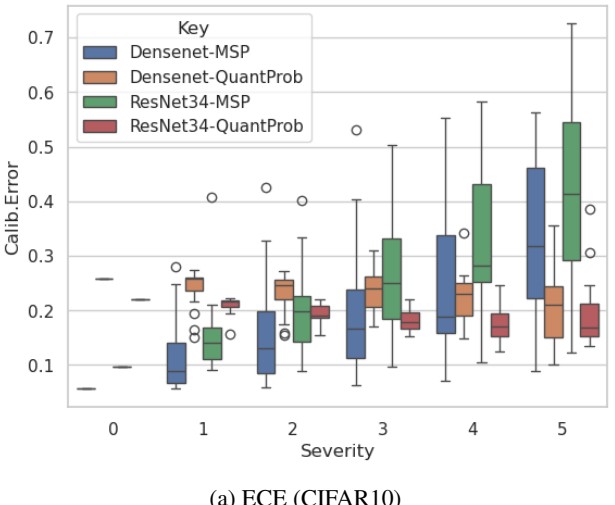

(a) ECE (CIFAR10)

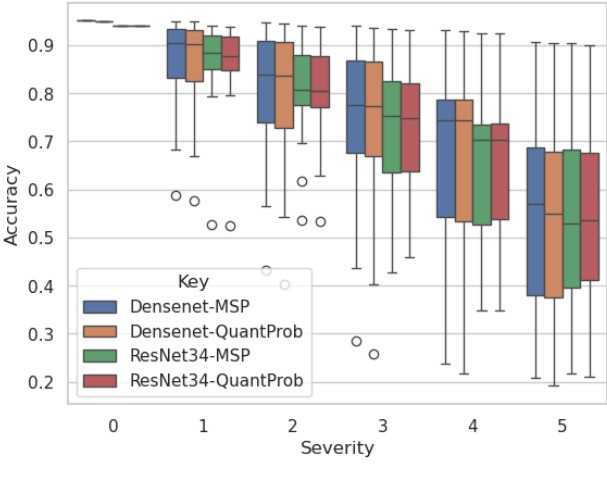

(b) Accuracy (CIFAR10)

Figure 2: Calibration errors when training on features from Resnet34/Densenet embedding on CIFAR10C. Quantile representations can be effective for calibration because they estimate probabilities using Equation equation 13, which has been shown to be robust to corruptions. As demonstrated using the CIFAR10C dataset [Hendrycks and Dietterich, 2019], the Expected Calibration Error (ECE) of the probabilities obtained from quantile representations (QUANT) does not increase with the severity of the corruptions. In contrast, when using the standard Maximum Softmax Probability (MSP) method, the calibration error increases as the severity of the corruptions increases.

approaches fail under distortions. In this section we show that QUANTPROB are robust to distortions.

Let $p_{i,k}$ denote the predicted probability that the sample $\boldsymbol{x}_i$ belongs to class $k$. A perfectly calibrated model (binary class) will satisfy [Guo et al., 2017] $P(\boldsymbol{y}_i = 1 | p_{i,1} = p^*) = p^*$. For multi-class case this is adapted to $P(\boldsymbol{y}_i = \arg\max_k(p_{i,k}) | \max_k(p_{i,k}) = p^*) = p^*$. The degree of mis-calibration is usually measured using *Expected Calibration Error (ECE)*

$$E[|p^* - E[P(\boldsymbol{y} = \arg\max_k(p_{i,k}) | \max_k(p_{i,k}) = p^*)]|]$$
(14)

This is computed by binning the probabilities into $m$ bins - $B_1, B_2, \cdots, B_m$ and computing $\hat{ECE} = \sum_{i=1}^{m}(|B_i|/n)|acc(B_i) - conf(B_i)|$. where $acc(B_i) = (1/|B_i|)\sum_{j\in B_i} I[\boldsymbol{y}_j = \arg\max_k(p_{j,k})]$ denotes the accuracy of the predictions lying in $B_i$, and $conf(B_i) = \sum_{j\in B_i}\max_k(p_{j,k})$ indicates the average confidence of the predictions lying in $B_i$. In practice, Kumar et al. [2019] proposes a better approach to estimate the *top-label* uncertainty which we use in this article.

**No Free Lunch for Calibration:** Is it possible to have an approach to assign probabilities which have constant calibration error across *all* probability distributions? The answer is unfortunately no.

This follows from a simple argument - Let $\mathcal{P}(X, Y)$ denote an underlying distribution of the samples where $Y \in \{0, 1\}$,

and let $f_\theta$ denote the model which is perfectly calibration for $\mathcal{P}(X, Y)$. Consider a new probability distribution $\mathcal{P}^+(X, Y) = \mathcal{P}(X, 1 - Y)$. Then the calibration error of $f_\theta$ on $\mathcal{P}^+(X, Y)$ is 0.5.

So, in general it is not possible to have constant calibration error across the entire space. The best one could hope for is to have constant calibration error whenever $\mathcal{P}^+(X, Y) \approx \mathcal{P}(X, Y)$, i.e invariant to small distortions. We show that QUANTPROB proposed in this article achieves this.

**Sanity Check - When the pre-tained model $f_\theta$ is perfect:** We firstly verify that, in the ideal scenario where the model is perfect, then the quantile probabilities match the perfectly calibrated probabilities. This is formalized in the theorem below.

**Theorem 4.1** *Let $f_\theta(.)$ denote the pre-trained model, and let $f_{\ell,\theta}(.)$ denote the corresponding logits. Assume that the data is generated using the model $\boldsymbol{y} = I[f_{\ell,\theta}(\boldsymbol{x}) + \epsilon > 0]$, where $\epsilon$ denotes the error distribution with mean 0 . Let $\mathcal{Q}(\boldsymbol{x}, \tau)$ denote the quantile representations obtained on this data using $f_\theta$ as the base classifier. Then,*

$$\int_{\tau=0}^{1} I[\mathcal{Q}(\boldsymbol{x}, \tau) \geq 0.5]d\tau = P(f_\theta(\boldsymbol{x}) + \epsilon \geq 0)$$
(15)

The proof for theorem 4.1 is given in the supplementary material. The main idea is the notion that $\mathcal{Q}(\boldsymbol{x}, \tau)$ captures $P(f_\theta(\boldsymbol{x}) + \epsilon > 1 - \tau)$.

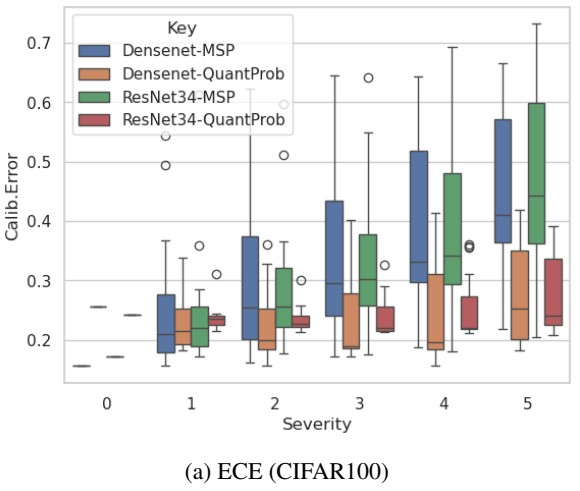

(a) ECE (CIFAR100)

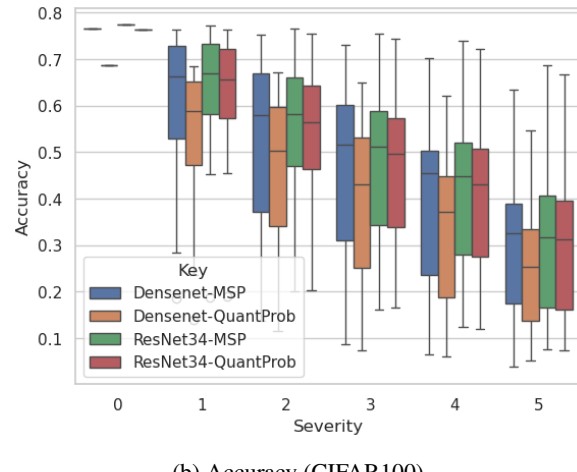

(b) Accuracy (CIFAR100)

Figure 3: Calibration errors when training on features from Resnet34/Densenet embedding on CIFAR100C

**When $f_\theta$ is not perfect:** Even in the case when the pre-trained model $f_\theta$ is not perfect, QUANTPROB generalizes better than the naive baseline $f_\theta(\boldsymbol{x})$. Figures 1c,1d provide evidence for this. Observe that the probabilities in figure 1c trace the manifold of the data distribution while the probabilities in figure 1d does not take into consideration the data distribution far away from the boundary. Thus, QUANTPROB on distorted distributions is much more reliable than the naive probabilities. We now empirically verify that QUANT-PROB generalize to a wider domain than the MSP in real world datasets.

**Experimental Setup** We verify that QUANTPROB generalize better than the naive probabilities by using a pre-trained ResNet34 on CIFAR10 dataset. To evaluate the QUANTPROB robustness to distortions, we use the `CIFAR10C` dataset introduced in Hendrycks and Dietterich [2019], which contains 15 types of common corruptions at five severity levels - $1, 2, 3, 4, 5$. The quantile-representations are obtained from the ResNet34 pre-trained on the CIFAR10 training data. We compare the performance with Maximum Softmax Probability (MSP) as a baseline and evaluate both accuracy and calibration error. To estimate calibration error, we construct the bins $\{B_i\}$ using 5 equally spaced quantiles within the predicted probabilities. The probabilities of each class are predicted using equation 13.

**Training on the features from the pretrained models:** Figure 2 presents how accuracy and calibration error varies with distortion. The main thing to observe is that - The calibration error of QUANTPROB remains constant across distortions while the usual MSP increases in the calibration error. Also note that while the standard deviation increases for both QUANTPROB and MSP, it increases quite drasti-

cally for MSP comparatively. Figure 2b verifies that this constant calibration error is not at the expense of reduction in accuracy.

**Training the entire networks:** Figure 4 shows the results when one trains the entire network instead of the last layer. We observe a similar trend - Calibration error QUANTPROB remains constant across distortions on average while MSP increases drastically.

This observation is interesting since - One might expect better results when training on the entire model instead of only on the features from last layer. Interestingly, we find that training the deep network does not improve the results. In fact we find that, while on average the calibration errors are similar, the standard deviation actually increases when compared to training only the last layer.

**Cannot Correct the Calibration Error Using Platt Scaling** Figure 2 shows that calibration error from quantile representations is approximately constant across distortions, but not zero. So – Does making the calibration zero on validation data make the calibration error zero across distortions? It turns out that usual methods fail when trying to correct the calibration error of quantile representations.

To verify this we perform the same experiment as earlier. Further we use Platt Scaling on validation data and accordingly transform the probability estimates for the corrupted datasets. These results are shown in figure 5. Observe that at severity 0, the calibration error is 0 for the corrected probabilities as expected. However, as distortion increases, the calibration error increases as well – a trend observed with using MSP probabilities.

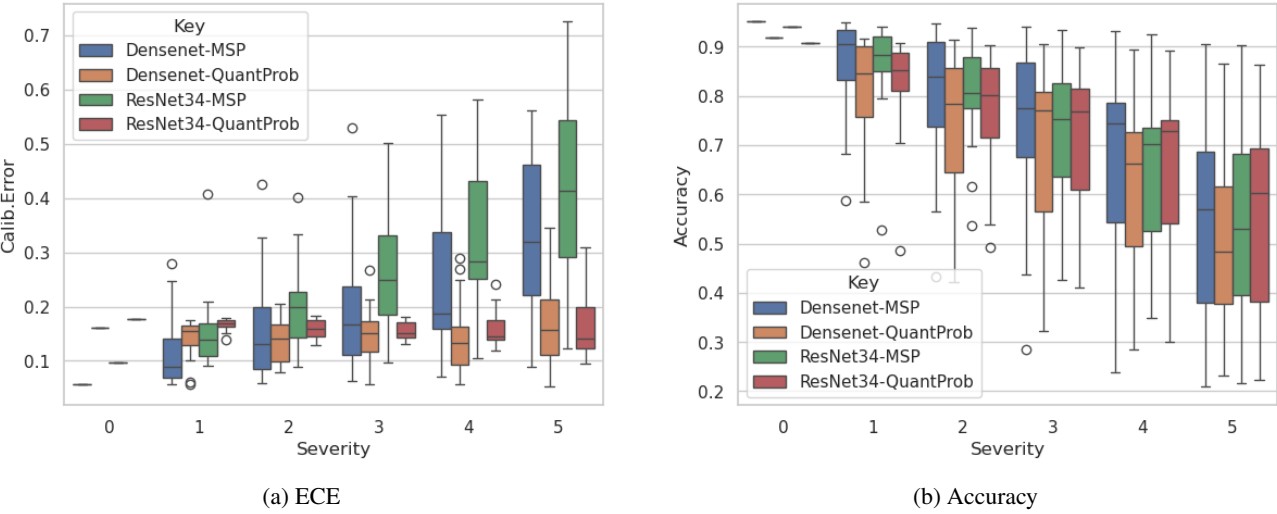

| (a) ECE | (b) Accuracy |

Figure 4: Calibration errors when training the entire network of Resnet34/DenseNet embedding on CIFAR10.

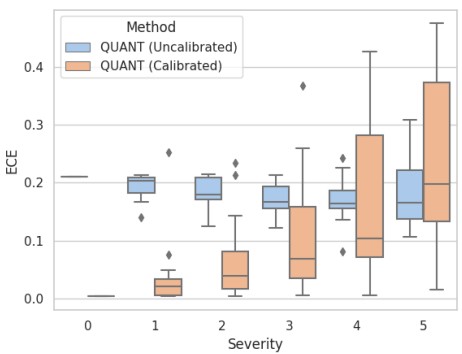

Figure 5: Correcting calibration error on the validation set may not improve performance on corrupted datasets.

## 5 RELATED WORK

[Koenker, 2005, Parzen, 2004, Portnoy and Koenker, 1989, Chaudhuri, 1992] provides a comprehensive overview of approaches related to quantile regression and identifying the parameters. [Chaudhuri, 1996] extends the quantiles to multi-variate case. [Tagasovska and Lopez-Paz, 2019, Tambwekar et al., 2022] use quantile regression based approaches for estimating confidence of neural networks based predictions. [Angelopoulos et al., 2021, Feldman et al., 2021b] uses conformal methods to calibrate probabilities, and is closely related to computing quantiles. [Chung et al., 2021] proposes a similar algorithm to overcome the restriction to pinball loss for regression problems. [Feldman et al., 2021a] generates predictive regions using quantile regression techniques.

## 6 CONCLUSION AND FUTURE WORK

**Summary:** Firstly, we argue that, from a systems perspective, it is more important to have constant calibration across distortions rather than minimal calibration error. The first is much more easier to correct by simply tuning the threshold, while the latter results in an unstable system. Having constant calibration error across distortions is also one of the open questions raised in Kumar et al. [2019].

The key issue which inhibits the current networks to have constant calibration across distortions is that - While networks are trained to generalize predictions, they are not trained to generalize probabilities. To correct this we resort to *quantile regression techniques*.

We aim to answer the question - *Given a pre-trained classifier $f_\theta$ with good performance, how can one assign the probabilities without changing the predictions?* We first establish a duality between quantiles and probabilities, and then use the duality to assign probabilities, QUANTPROB which generalize better. We then show that QUANTPROB results in a calibration error which is constant across distortions while the usual MSP increases the calibration error drastically.

**Open Questions:** The ideal scenario is, of course, having minimal calibration error across all distributions. However, we have a no-free-lunch result and hence one cannot have constant calibration across all distributions. We observed that any attempts at correcting the calibration error, either by using a larger networks or by using traditional approaches like Platt-scaling, resulted in increasing either the standard deviation across distortions or increasing the calibration error itself across distortions.

Thus, while in this article we achieve constant calibration error across distortions, "How to obtain minimal calibration error across distortions?" remains an open question. We wish to pursue this as future work.

**Related Applications and Analysis:** Apart from the application to calibration of probabilities, QUANTPROB can also be used for OOD detection. In fact we find that QUANT-PROB behaves similarly to the recent state-of-the-art MLS approach [Vaze et al., 2022]. Details about the experiments can be found in appendix C. Apart from that, we also show that quantile representations capture the distribution of the dataset by considering cross-correlations. This can be found in appendix E. We also empirically confirm that the quantile representations preserve monotonicity in appendix G.

# 7 ACKNOWLEDGEMENT

Aditya Challa acknowledges the support from CEFIPRA(68T05-1) . Snehanshu Saha and Aditya Challa would like to thank the Anuradha and Prashanth Palakurthi Center for Artificial Intelligence Research (APPCAIR) and SERB CRG-DST (CRG/2023/003210) for support. Snehanshu Saha acknowledges SERB SURE-DST (SUR/2022/001965) and the DBT-Builder project (BT/INF/22/SP42543/2021), Govt. of India for partial support.

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

# QUANTPROB: Generalizing Probabilities along with Predictions for a Pre-trained Classifier
# (Supplementary Material)

**Aditya Challa**[1]                **Snehanshu Saha**[1]                **Soma S. Dhavala**[2]

[1]Department of CSIS and APPCAIR , Birla Institute of Technology and Science, Goa, India ,
[2]Director-ML, Wadhwani AI, Bengaluru, India ,

## A   PROOF FOR THEOREM 3.1

Let $\mathcal{D} = \{(\boldsymbol{x}_i, y_i)\}$ denote the train dataset of size $N$. Recall, equation 11 (main article) is

$$\min_{\psi} \mathbb{E}_{\tau \sim U[0,1]} \left[ \frac{1}{N} \sum_i \rho(I[\psi(\boldsymbol{x}_i, \tau) \geq 0.5], y_i; \tau) \right] \tag{16}$$

Let $\mathcal{Q}(\boldsymbol{x}, \tau)$ denotes the solution obtained using the algorithm 1 (main article). Let $\Phi(\boldsymbol{x}, \tau)$ denote the solution obtained by solving equation 16.

**Aim** to show that $I[\mathcal{Q}(\boldsymbol{x}_i, \tau) \geq 0.5] = I[\Phi(\boldsymbol{x}_i, \tau) \geq 0.5]$ for all the points in $\mathcal{D} = \{(\boldsymbol{x}_i, y_i)\}$, for all $\tau$.

From construction in the algorithm 1,we have $\mathcal{Q}(\boldsymbol{x}_i, \tau) = I[\Phi(\boldsymbol{x}_i, 0.5) \geq 1 - \tau]$, since $\Phi(\boldsymbol{x}_i, 0.5)$ is considered as the base classifier.

Now, under the assumption of strong duality, if $\Phi(\boldsymbol{x}_i, 0.5) = 1 - \tau$, then for all quantiles $\tau^* \geq \tau$, $\Phi(\boldsymbol{x}_i, \tau^*) \geq 0.5$ (see section 3 above). Hence, $\mathcal{Q}(\boldsymbol{x}_i, \overline{\tau) = I[\Phi(\boldsymbol{x}_i}, \tau) \geq 0.5]$. This implies that, $I[\mathcal{Q}(\boldsymbol{x}_i, \tau)] \geq 0.5] = I[\Phi(\boldsymbol{x}_i, \tau) \geq 0.5]$.

## B   PROOF FOR THEOREM 4.1

From the construction of $\mathcal{Q}(\boldsymbol{x}, \tau)$

$$I[\mathcal{Q}(\boldsymbol{x}, \tau) \geq 0.5] \Leftrightarrow I[f_\theta(\boldsymbol{x}) \geq (1 - \tau)] \Leftrightarrow P(f_{\ell,\theta}(\boldsymbol{x}) + \epsilon \geq 0) \geq 1 - \tau \tag{17}$$

The second equality follows from the assumption that $f_\theta(x_i)$ denotes the proportion of times we observe $y_i = 1$ given $x = x_i$. This holds true for any well-trained classifier. Assuming that $\tau^* = P(f_{\ell,\theta}(\boldsymbol{x}_i) + \epsilon \geq 0)$, So, we have

$$\begin{aligned}
\int_{\tau=0}^{1} I[\mathcal{Q}(\boldsymbol{x}_i, \tau) \geq 0.5] d\tau &= \int_{\tau=0}^{1} I[\tau^* \geq (1 - \tau)] d\tau \\
&= \int_{\tau=0}^{1} I[\tau \geq (1 - \tau^*)] d\tau = \int_{\tau=(1-\tau^*)}^{1} 1 d\tau = \tau^*
\end{aligned} \tag{18}$$

Thus the theorem follows.

## C   OOD DETECTION USING QUANTPROB

An assumption made across all machine learning models is that - Train and test datasets share the same distributions. However, test data can contain samples which are out-of-distribution (OOD) whose labels have not been seen during the training process [Nguyen et al., 2015]. Such samples should be ignored during inference. Hence OOD detection is a key

Table 1: Comparison of Quantile-Representations with baseline for OOD Detection.The entries are represented as `MQP/MLS/MSP`. Observe that except in a few cases, `MQP` and `MLS` perform comparably for OOD detection.

| Model/ID | OOD | AUROC | TNR-TPR95 | Det.Acc |
|---|---|---|---|---|
| ResNet34/CIFAR10 | Imagenet(C) | **92.68**/92.27/90.96 | **63.88**/57.31/43.95 | **86.26**/85.70/84.80 |
| | Imagenet(R) | **92.13**/91.47/90.33 | **61.44**/53.37/42.18 | **85.69**/84.90/84.21 |
| | LSUN(C) | 92.51/**93.53**/91.74 | **64.00**/61.77/45.87 | 86.92/**87.24**/86.37 |
| | LSUN(R) | **94.83**/91.55/90.07 | **71.87**/55.07/41.24 | **88.96**/85.20/84.25 |
| | iSUN | **94.17**/91.76/90.29 | **70.36**/55.74/41.90 | **87.96**/85.27/84.28 |
| Resnet34/SVHN | Imagenet(C) | **94.34**/93.75/94.18 | 82.34/**82.86**/81.13 | 89.14/90.97/**91.23** |
| | Imagenet(R) | **93.53**/92.89/93.52 | 80.67/**81.44**/79.86 | 88.29/90.17/**90.58** |
| | LSUN(C) | 88.93/92.59/**92.99** | 68.54/**79.82**/77.96 | 83.59/89.68/**90.10** |
| | LSUN(R) | 90.69/90.53/**91.50** | 72.86/**76.82**/74.95 | 84.72/88.56/**89.09** |
| | iSUN | 91.23/91.50/**92.28** | 74.76/**79.45**/77.43 | 85.73/89.26/**89.77** |

component of reliable ML systems. Several methods [Hendrycks and Gimpel, 2017, Lee et al., 2018, Bibas et al., 2021] have been proposed for OOD detection.

Intuitively, OOD samples are far from the boundary and result in low softmax probabilities. Thus, one way to assign OOD scores to samples is by considering the maximum softmax probabilities (`MSP`) as described in [Hendrycks and Dietterich, 2019]. Samples which are far from the boundary also have large logit scores. In Vaze et al. [2022] the authors suggest to use maximum logit score (`MLS`) instead and show that this is indeed a state-of-the-art approach for identifying OOD samples.

To assign an OOD score for the quantile representations we use *maximum quantile probabilities* (`MQP`) over all the classes, that is, if $p_{i,k}$ denotes the quantile probability obtained using equation 13 of sample $i$ belonging to class $k$, then

$$\texttt{MQP}(\boldsymbol{x}_i) = \max_k \{p_{i,k}\}$$

**Relation between `MQP` and `MLS`:** Another interpretation of QUANTPROB in equation 13 is that it measures the distance (in terms of quantiles) from the boundary. If $p_{i,k} = 1$, then $\mathcal{Q}_\ell(\boldsymbol{x}_i, \tau) \geq 0$ for all $\tau$, which implies $f_{\ell,\theta}(\boldsymbol{x}_i)$ is larger than $(1 - \tau)$ quantile of $\{f_{\ell,\theta}(\boldsymbol{x}_j)\}_j$ for all $\tau$. Thus, the $f_{\ell,\theta}(\boldsymbol{x}_i)$ has a high latent score which implies high `MLS` score. Similar argument holds for low latent scores as well. (**Remark:** This is evident in the illustration in figure 1b.) Thus, `MQP` and `MLS` perform similarly for OOD detection. We verify this below.

**Experimental Setup:** For this study, we use the CIFAR10[Krizhevsky et al., 2014] and SVHN[Netzer et al., 2011] datasets as in-distribution (ID) datasets and the iSUN[Xu et al., 2015], LSUN[Yu et al., 2015], and TinyImagenet[Liang et al., 2018] datasets as out-of-distribution (OOD) datasets. Two versions of LSUN and TinyImagenet are considered - resized to $32 \times 32$ and cropped. We evaluate the quantile representations obtained using ResNet34[He et al., 2016] architecture. For evaluation we use (i) AUROC: The area under the receiver operating characteristic curve of a threshold-based detector. A perfect detector corresponds to an AUROC score of 100%. (ii) TNR at 95% TPR: The probability that an OOD sample is correctly identified (classified as negative) when the true positive rate equals 95%. (iii) Detection accuracy: Measures the maximum possible classification accuracy over all possible thresholds.

**Results:** Table 1 shows the results comparing `MQP`, `MLS` and `MSP`. As argued before, `MQP` and `MLS` perform similarly in comparison with `MSP`.

# D   RESULTS WHEN TRAINING ONLY THE LAST LAYER

The same observations as done in the main article also hold true when training is done only in the last layer by considering the features.

**OOD Detection :** These experiments were performed using Densenet and Resnet34 architectures on CIFAR10 and SVHN datasets. The OOD datasets are the same as in the main article. Table 2 shows the results obtained when quantile representations are used only on the last layer.

Table 2: Comparison of Quantile-Representations with baseline for OOD Detection. Observe that Quantile-Representations outperform the baseline in all the cases.

| | | DenseNet (Baseline/Quantile-Rep) | | | | |
|---|---|---|---|---|---|---|
| | | LSUN(C) | LSUN(R) | iSUN | Imagenet(C) | Imagenet(R) |
| | AUROC | 92.08/93.64 | 93.86/94.61 | 92.84/93.74 | 90.93/91.72 | 90.93/92.06 |
| CIFAR10 | TNR@TPR95 | 58.19/64.56 | 63.07/66.89 | 59.64/64.68 | 53.94/56.34 | 54.44/58.22 |
| | Det. Acc | 85.58/87.14 | 87.66/88.60 | 86.29/87.42 | 84.11/84.93 | 84.10/85.33 |
| | AUROC | 91.80/92.29 | 90.75/90.70 | 91.21/91.30 | 91.93/91.97 | 91.93/92.01 |
| SVHN | TNR@TPR95 | 54.61/58.77 | 47.67/48.55 | 48.24/50.15 | 52.38/53.68 | 52.43/53.64 |
| | Det. Acc | 85.10/85.37 | 84.32/84.16 | 84.80/84.77 | 85.42/85.55 | 85.46/85.50 |
| | | Resnet34 (Baseline/Quantile-Rep) | | | | |
| | AUROC | 91.43/91.76 | 92.64/93.08 | 91.89/92.34 | 90.59/90.81 | 89.12/89.39 |
| CIFAR10 | TNR@TPR95 | 54.96/56.76 | 63.24/65.75 | 58.56/60.94 | 52.86/54.89 | 47.41/49.93 |
| | Det. Acc | 84.63/84.96 | 85.41/86.06 | 84.39/85.17 | 83.24/83.44 | 81.74/82.05 |
| | AUROC | 94.80/94.87 | 94.37/94.46 | 95.13/95.22 | 95.73/95.85 | 95.62/95.70 |
| SVHN | TNR@TPR95 | 76.19/76.15 | 72.10/72.87 | 75.88/76.25 | 79.16/79.53 | 78.34/78.82 |
| | Det. Acc | 89.58/89.72 | 88.82/88.87 | 89.78/89.85 | 90.72/90.87 | 90.54/90.60 |

**Calibration Experiments** The same observations - Quantile probabilities have calibration error which does not change with distortion and that these could not be corrected using simple Platt Scaling/Isotonic Regression, hold true when training only the last layer as well. This is illustrated in figure 8.

## E ANALYSIS OF CROSS-CORRELATION

To illustrate that the quantile representations capture the aspects of data-distrbution relevant to classification, we perform the following experiment - Construct the cross-correlation between features using (i) Quantile Representations and (ii) Feature values extracted using the traindata. If our hypothesis is accurate, then cross-correlations obtained using quantile-representations and feature values would be similar.

In Figures 6 and 7, we present the results of using features from Resnet34 and Densenet on the CIFAR10 dataset. Figures 6a and 6b show the results for Resnet34, and Figures 7a and 7a show the results for Densenet. To visualize the cross-correlations, we use a heatmap with row and column indices obtained by averaging the linkage of train features. This index is common for both quantile representations and extracted features. It is evident from the figure that the cross-correlation between features is similar whether it is computed using extracted features or quantile representations.

## F A CASE WHERE QUANTILE REPRESENTATIONS DO NOT CAPTURE THE ENTIRE DISTRIBUTION

In figure 9 we illustrate an example where quantile representations do not capture the entire distribution. Here we use the same data as in figure 1, but with different class labels. This is shown in figure 9a. When we perform the OOD detection we get the region as in figure 9b. Observe that while it does detect points far away from the data as out-of-distribution, the moon structure is not identified. In particular, the spaces between the moons is not considered OOD. This illustrates a case when quantile representations might fail.

However, OOD detection using a single classifier also fail, as illustrated in figure 9c. Observe that the region identified by quantile representations is much better than the one obtained using a single classifier.

**A simple fix for OOD detection:** If OOD detection were the aim, then it is possible to change the approach slightly by considering *random labels* instead of the ground-truth labels. This allows us to identify arbitrary regions where the data is located. This is illustrated in figure 9d. Observe that this method can be used to identify any region in the space by suitably sampling and assigning pseudo-labels. In this case, we identify the training data perfectly.

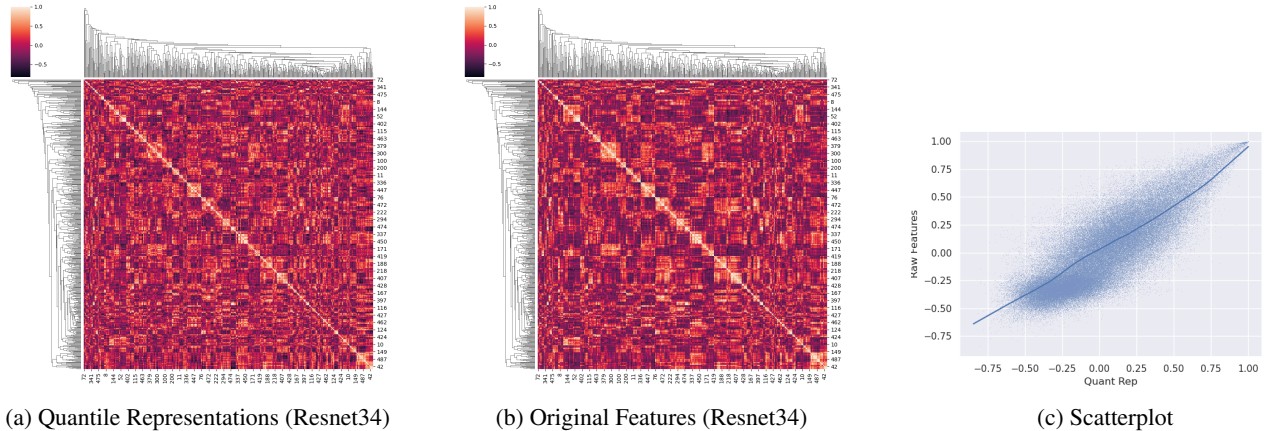

|  (a) Quantile Representations (Resnet34) | (b) Original Features (Resnet34) | (c) Scatterplot |

Figure 6: Do quantile representations capture the relevant information for classification? (a) Cross-correlations obtained using Quantile representations for Resnet34 on CIFAR10 (b) Cross-correlations obtained using train features for Resnet34 on CIFAR10. (c) Scatterplot with best fit line (using Locally Weighted Scatterplot Smoothing[Cleveland, 1979]) of the cross-correlation of features. Observe that as the correlation becomes important (i.e close to $-1$ or $1$) quantile representations are more consistent with raw features.

## G SANITY CHECK - PRESERVING MONOTONICITY PROPERTY

Note that quantile representations obtained by optimizing the simulateneous loss equation 2, should follow the monotonicity property - $\mathcal{Q}(\boldsymbol{x}, \tau_0) \leq \mathcal{Q}(\boldsymbol{x}, \tau_1) \leftrightarrow \tau_0 \leq \tau_1$. Since our approach is an alternate, the quantile representation learnt using algorithm 1 should satisfy this property as well. We verify this as follows - Considering the ResNet34 architecture trained on CIFAR10 dataset, we plot the *logits* obtained at different quantiles.

## H TRAINING DETAILS AND COMPUTE

Code is provided at `https://github.com/adityac20/quantprob.git` for more details about the exact training.

Training quantile representations was done on a DGX server using 4 GPUs. Observe that technically the size of the dataset increases by number of quantiles for training. However, starting from the pre-trained weights, using Adam optimizer with learning rate $3e-4$, we found that the network converges fairly quickly after 10-15 epochs. On the DGX server with 4 GPUs, it takes around 4 hours to reach convergence.

## I WHY QUANTILE REGRESSION?

If the goal of a regression problem is to predict the likely range of estimates (prediction interval) and not just a single estimate as the Ordinary Least Square Regression (OLS) does, the method is required to be more general and robust. This method for producing such estimates, relatively unknown in the Machine Learning community, is known as quantile regression. While OLS regression minimizes the squared-error loss function to predict a single point estimate, quantile regressions minimize the quantile loss in predicting a certain quantile. The 50th percentile, otherwise known as the median, represents the quantile loss as the sum of absolute errors (MAE). Other quantiles could give endpoints of a prediction interval; for example, a middle-80-percent range is defined by the 10th and 90th percentiles. The quantile loss differs depending on the evaluated quantile, such that more negative errors are penalized more for higher quantiles and more positive errors are penalized more for lower quantiles. In other words, quantile loss varies with the error, depending on the quantile, commonly interpreted as quantile for under- and over-estimated predictions. The higher the quantile, the more the quantile loss function penalizes underestimates and the less it penalizes overestimates. Quantiles allow for an understanding of a probability distribution of a data set in which only the specifications of the positions are known. Thus, wherever predictions are subject to high uncertainty, quantile should be the preferred loss function. Quantiles give some information about the shape of a distribution - in particular whether a distribution is skewed or not; are robust to outliers and can model extreme events well. Conditional quantiles obtained via regression are used as a robust alternative to classical conditional means in

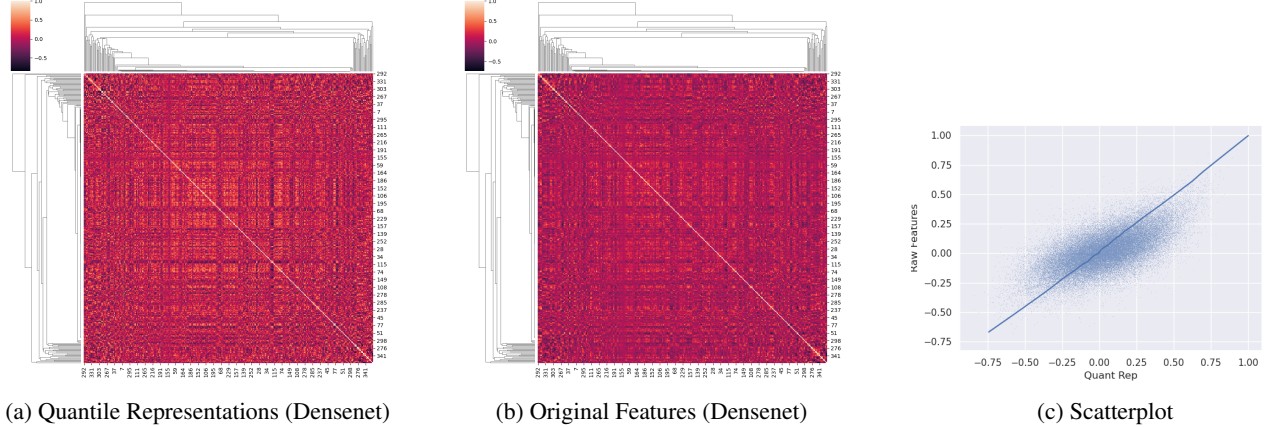

| (a) Quantile Representations (Densenet) | (b) Original Features (Densenet) | (c) Scatterplot |

Figure 7: Do quantile representations capture the relevant information for classification? (a) Cross-correlations obtained using Quantile representations for Densenet on CIFAR10 (b) Cross-correlations obtained using train features for Densenet on CIFAR10. (c) Scatterplot with best fit line (using Locally Weighted Scatterplot Smoothing[Cleveland, 1979]) of the cross-correlations. Observe that as the correlation becomes important (i.e close to $-1$ or $1$) quantile representations are more consistent with raw features.

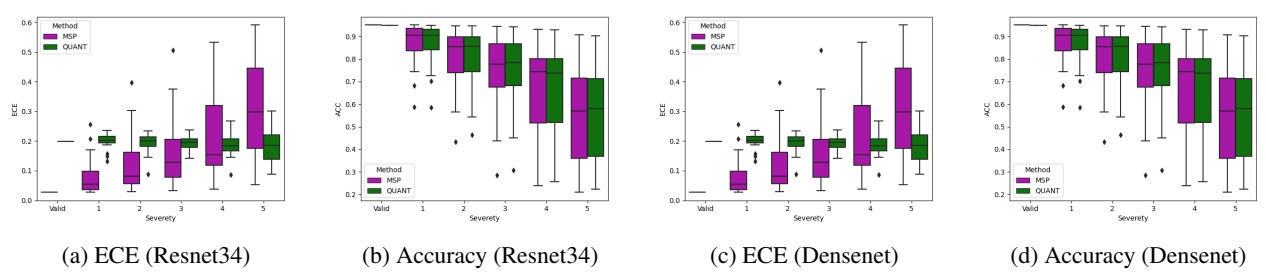

| (a) ECE (Resnet34) | (b) Accuracy (Resnet34) | (c) ECE (Densenet) | (d) Accuracy (Densenet) |

Figure 8: Quantile representations can be effective for calibration because they estimate probabilities using Equation equation 13, which has been shown to be robust to corruptions. As demonstrated using the CIFAR10C dataset [Hendrycks and Dietterich, 2019], the Expected Calibration Error (`ECE`) of the probabilities obtained from quantile representations (`QUANT`) does not increase with the severity of the corruptions. In contrast, when using the standard Maximum Softmax Probability (`MSP`) method, the calibration error increases as the severity of the corruptions increases.

econometrics and statistics, as they can capture the uncertainty in a prediction, and model tail behaviors, while making very few distributional assumptions

The quantile regression has started relatively recently being applied in the energy-growth nexus literature. In the past, it has been used extensively in pediatric medicine (offering an optimistic perspective for precision medicine), survival and duration time studies [Huang et al., 2017], the determination of wages, discrimination effects, and income inequality. Also, it has been used in the finance literature in studies that dealt with bank failure and the time occurrence of this failure [Schaeck, 2008]. Regarding the more recent application in the energy-growth nexus field, it is not well documented in the relevant studies why asymmetries would be present in the way income and wealth is generated in different countries given the consumption of energy in those countries and other stylized parameters. One reason, quite understandable, why to use this method, is for testing whether poorer countries will be affected the same way by energy conservation measures as the rich ones. Another reason as stated by Troster et al. [2018] in their study on renewable energy, oil prices, and economic growth for the United States is that their study would allow them to determine whether extremely low or high changes in energy consumption prices would lead economic growth. Therefore we can have very specific and accurate answers to what will happen if there is 1% energy reduction in poor countries. This information would otherwise have to be included in dummy variables and other forms of robust estimation that assign less weight to observations that are characterized as outliers. Among the various other statistical twists offered by the method, the quantile regression may be favored because it does not

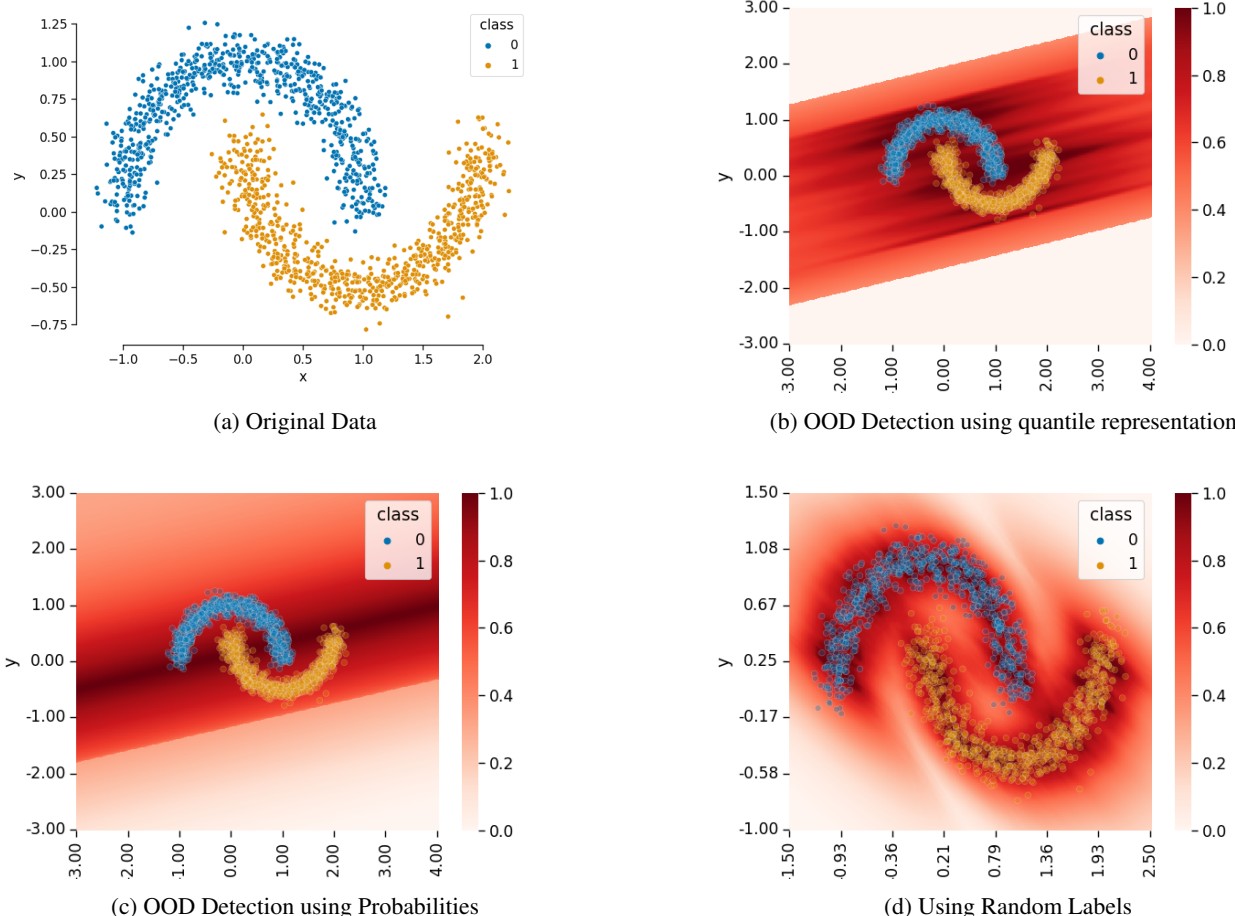

(a) Original Data

(b) OOD Detection using quantile representations

(c) OOD Detection using Probabilities

(d) Using Random Labels

Figure 9: Illustrating a case where quantile representations do not capture the distribution perfectly. (a) Original Dataset. (b) The region detected as in-distribution by using quantile representations. (c) Region detected as in-distribution by using the outputs from a single classifier. Observe that quantile representations still perform better than single classifier outputs. (d) Using random labels instead of ground-truth. Observe that the two moons structure is faithfully preserved in this image. The brightness of Red indicates the chance of being in-distribution.

assume a parametric distribution and it estimates the entire conditional distribution of the independent variable. Generally, this method is regarded as more versatile and informative [Rodriguez and Yao, 2017].

A switch from the squared error to the tilted absolute value loss function allows gradient descent-based learning algorithms to learn a specified quantile instead of the mean. It means that we can apply all neural network and deep learning algorithms to quantile regression [Huang et al., 2017, Schaeck, 2008]. The application of quantiles in deep learning, although relatively recent, are critical for model interpretability. In the past, [Tambwekar et al., 2022] extended the notion of conditional quantiles to the binary classification setting—allowing uncertainty quantification in the predictions, increased resilience to label noise thus furnishing new insights into the functions learnt by the models. This was accomplished by defining a new loss called binary quantile regression loss, in the classification setting. The estimated quantiles to obtain individualized confidence scores provide an accurate measure of a prediction being misclassified. These scores were then aggregated to compute two additional metrics, namely, confidence score and retention rate, which can be used to withhold decisions and increase model accuracy. Thus, in a non-parametric binary quantile classification framework, authors could demonstrate that quantiles aid in explainability as they can be used to obtain several uni-variate summary statistics that can be directly applied to existing explanation tools.

Therefore, it is not unconvincing to realize the relevance and precedence of quantiles in classification, in particular, to obtain the conditional quantiles of the underlying latent function learnt by a binary classifier using customized loss inspired by

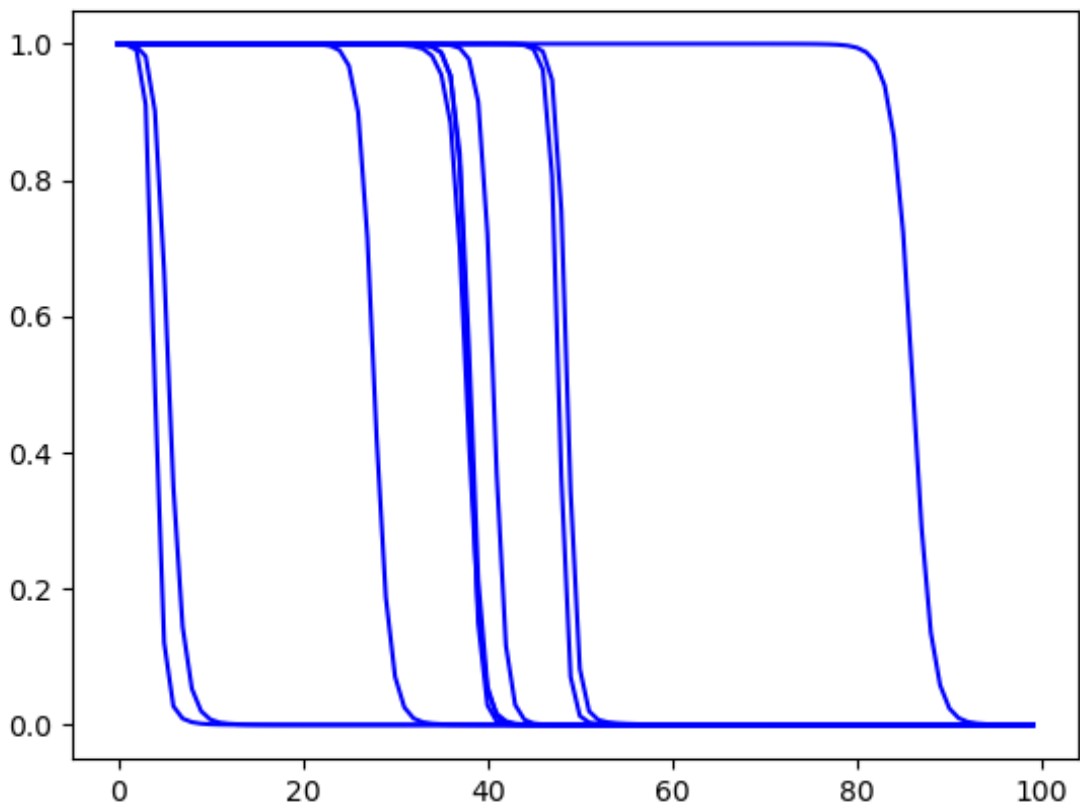

Figure 10: Checking that the quantile representations learnt using algorithm 1 satisfies the monotonicity property.

quantiles [Troster et al., 2018].