# OpenReview forum: "QuantProb: Generalizing Probabilities along with Predictions for a Pre-trained Classifier"
_auai.org/UAI/2024/Conference — UAI 2024 poster_

### Official Review · Reviewer_Ed8t · 2024-03-20

**Q2-1 Originality-Novelty:** 3
**Q2-2 Correctness-Technical Quality:** 3
**Q2-5 Clarity Of Writing:** 4

**Q1 Summary And Contributions:**

In this paper the authors propose a method called QuantProb, with the purpose of improving model calibration in classification. In the method, the authors take a pre-trained classifier and retrain the classifier with quantile regression. One of the main contributions is decoupling the construction of quantile representations from the loss functions, which the authors argue ensures that the classifier does not lose any of the already trained properties. The authors show that in contrast to softmax classifiers, the models retrained with QuantProb do not suffer from changing calibration errors under increasingly severe distribution shifts.

**Q2-3 Extent To Which Claims Are Supported By Evidence:**

2: Fair: the main claims are somewhat supported by evidence (but the experimental evaluation may be weak, or does not match entirely with the claims, important baselines may be missing, proofs contain important ideas but lack rigor, algorithmic details are only discussed superficially, references are imprecise, assumptions are not sufficiently motivated or explicated, etc.).

**Q2-4 Reproducibility:**

2: Fair: key resources (e.g. proofs, code, data) are unavailable but key details (e.g. proof sketches, experimental setup) are sufficiently well-described for an expert to confidently reproduce the main results.

**Q3 Main Strengths:**

* The paper is very well written and organized.
* The method is, to my knowledge, an interesting and new direction for how to create robust neural network classifiers.

**Q4 Main Weakness:**

* In the experimental section, the authors only showcase their method on CIFAR10 and CIFAR10C, leaving the empirical evidence somewhat lacking in my opinion.
* The authors provide some very strong statements regarding calibration and neural network training which I unfortunately do not think is completely true. I will elaborate further below.

**Q5 Detailed Comments To The Authors:**

1. Regarding the experimental showcase, I think the paper would significantly benefit from more experimental evidence in the main body text. Perhaps the authors could run CIFAR100 and CIFAR100C and run the same experiments, or as a minimum move some of the experiments from the supplementary material to the main text?
2. Regarding statements on calibration and neural network training, I have the two following comments:
    * Firstly, I disagree with the current statement in the paper that _it is more crucial to have calibration errors which do not change with distortions, rather than small calibration errors_. Although I agree with the sentiment that it is of course desirable to have constant calibration error across distortions, I think that small calibration errors are equally important. In the extreme case, a model that has very high calibration error but constant under distortions still fits under this statement. Perhaps the authors could amend the statement to be less strong? In addition, I am curious why the authors necessarily think that being constant and having low ECE is necessarily at contrast with each other?
    * Secondly, to my understanding the statement _while networks are trained to generalize predictions, they are not trained to generalize probabilities_ is also incorrect when training models for classification using log likelihood loss. We know this because log likelihood is a proper scoring rule, and thus as stated in https://probml.github.io/pml-book/book2.html (Page 572), _Therefore minimizing the NLL (aka log loss) should result in well-calibrated probabilities._. Perhaps the authors could elaborate on their statement or argue why they think it is true?
3. As seen in Figures 2 and 3, the ECE of the QuantProb trained models is significantly lower than for the base models at severity 0 although the accuracies are the same. Could the authors perhaps comment on this? I am just curious to understand why this might be the case.


* Minor Comments:
    * In Figures 2 and 3, the colors of the different models in the legend swap around which can be a little bit confusing in my opinion.
    * Is there a reason the authors use 5 bins in their ECE evaluations rather than a larger number? 10 or 15 is more commonly used to the best of my knowledge.

------------------------------------------------------------------

POST DISCUSSION EDIT

The authors have addressed my main weaknesses (in particular those on strong statement) and provided additional experimental results which support the initial findings. For this reason I would like to raise my score to a 6 (previously a 4).

------------------------------------------------------------------

**Q9 Complying With Reviewing Instructions:**

Yes

---

> ### Author Rebuttal · Authors · 2024-04-02
>
> **"it is more crucial to have calibration errors which do not change with distortions, rather than small calibration error"*** - We would like to still argue that this is the case. We agree with the reviewer that the ideal scenario is $0$ calibration error across distortions. However, given a choice between - (a) increasing calibration error across distortions, with $0$ calibration at severity=0 or (b) constant (non zero) calibration error across distortions -- we want to argue that (b) is a better option.
>
> Consider a hypothetical case where a ML system is switched to a human reviewer based on the threshold of the probability. If the probabilities are "predictably" going wrong with small distortions (which happens when calibration errors are constant across distortions) - this is much easier to correct (say) by setting an appropriate threshold of switching and we expect that these thresholds are going to be consistent across distortions. On the other hand, if the probabilities have very large calibration errors under distortions, then such thresholds might not be possible. While this is an hypothetical case, as a philosophy we would like to prefer "predictably" going wrong.
>
> Another argument for our position is - Having models which can ensure calibration error constant, restrict the number of possible output distributions the models can potentially take. This is (in our opinion) a natural inductive bias (for models aiming to be distribution independent) which existing models do not exploit/use. A restriction on the possible output distributions can allow us to tackle robustness issues.
>
> However, we do understand that this may not be widely held stance and will include a discussion on this aspect in the revision and amend the sentences accordingly.
>
> **"while networks are trained to generalize predictions, they are not trained to generalize probabilities"** We do not imply that NLL probabilities are not well calibrated, but rather if a model is trained on distribution $X$, the probabilities are usually uncalibrated when the distribution changes slightly (say) $X \to X^+$. We meant that the current training loss/architectures do not take distortions into consideration when training. This is what we are trying to correct with the framework proposed in this article.
>
> **Experiments on CIFAR100C-** Please find the results for CIFAR100C (training only the last layer) at [accuracy](https://anonymous.4open.science/r/quantprob-C207/reports/boxplot_accuracy_cifar100.png) and [calibration error](https://anonymous.4open.science/r/quantprob-C207/reports/boxplot_calib_error_cifar100.png). Note that the main observations do not change here as well.
>
> **ECE of the QuantProb trained models is significantly lower than for the base models** The reasoning we offer in the article is as follows:
>
> 1. Under distortions while classifiers can still predict the labels, the probabilities have bad calibration.
> 2. We convert the probability assignment problem to a sequence of classification problems using duality and the quantile framework in the article. If the classifiers can predict the label well under distortions, then taking an average of these predictions give a probability which is robust to distortions.
> 3. Further we have theorem 4.1. Let $X \to X^{+}$ denote a arbitrary distortion. If (slightly moving away from notation in the article) the labels are given by $\hat{Y}(X) = I[f_{\theta}(X) + \epsilon \geq 0]$, then we know that $(X, \hat{Y}(X))$ and $(X^+,\hat{Y}(X^+))$ both have calibration error $0$ under $f_{\theta}$. This has a "normalizing effect". Now, when these distributions change as $(X, \hat{Y}(X)) \to (X, Y)$ (Y denotes the actual gt-labels), and $(X^{+}, \hat{Y}(X^{+})) \to (X^{+}, Y)$, observe that it is only the labels which change. So, we expect that calibration error differences between these two shifts are approximately the same under small distortions. Hence, on an average we expect the calibration error to remain constant.
>
> For a more concrete intuition consider an hypothetical extension to figure 1 in the article. In [img1](https://anonymous.4open.science/r/quantprob-C207/img/Figure_scatter.png) we distort the dataset slightly by adding translations and compute the calibration errors in [img2](https://anonymous.4open.science/r/quantprob-C207/img/Figure_calib.png). The same behaviour as observed in CIFAR10C is repeated here as well.
>
> Do note that this behaviour is only for a combination of meaningful classifiers and meaningful distortions. In general, because of no-free-lunch in calibration, we cannot provide rigorous assurances. Unless, we know reasonable assumptions to make on the possible set of distortions.

---

### Official Review · Reviewer_KpL5 · 2024-03-20

**Q2-1 Originality-Novelty:** 2
**Q2-2 Correctness-Technical Quality:** 3
**Q2-5 Clarity Of Writing:** 2

**Q1 Summary And Contributions:**

The paper presents an approach for obtaining calibrated probabilities from the outputs of a pretrained classifier. The authors propose to exploit quantile regression for this purpose; more particularly, they exploit a duality property of the pinball loss to derive a so-called duality property between quantiles and probabilities. The input of the strategy is a classifier providing class probabilities (logits). The original dataset can be augmented using a binary information indicating, for each instance, whether the $\tau$th quantile of the logits is exceeded or not. A new probabilistic (logit) predictor can then be trained on this augmented dataset, for all quantile values. Experiments are realized on the CIFAR-10 dataset to show that the proposal generalizes better than the base probabilities (provided by the original classifier), studying also the robustness of the model to distortions on the CIFAR10C variant.

Section 1 (Introduction) presents the setting. Section 2 discusses a related problem, simultaneous binary quantile regression. Section 3 presents the proposal: first, the "duality property" between quantiles and probabilities is presented; then, the problem of generating quantile representations in practice is addressed. Section 4 studies how the proposal can be applied to obtained calibrated decisions. Section 5 discusses related works, before a conclusion is drawn in Section 6.

**Q2-3 Extent To Which Claims Are Supported By Evidence:**

1: Poor: the authors fail to convincingly backup their main claims (e.g., if the experimental evaluation is flawed, proofs are lacking or invalid, references are missing, assumptions are not realistic, not specified, or not motivated).

**Q2-4 Reproducibility:**

3: Good: key resources (e.g. proofs, code, data) are available and key details (e.g. proofs, experimental setup) are sufficiently well-described for competent researchers to confidently reproduce the main results.

**Q3 Main Strengths:**

The topic (calibration of probabilistic classifiers) considered in the paper is interesting.

The proposal ingeniously leverages a property of the pinball loss to address the issue.

**Q4 Main Weakness:**

The proposal, although sound, does not seem to offer strong calibration guarantees. Theorem 3.1 seems to rely on a strong duality property, the satisfiability of which is not clear. Theorem 4.1 proves the optimality of the approach when the base classifier is perfect, which is arguably a very strong assumption.

The approach seems restricted to binary classifiers. The extension to the multiclass case is not clear.

The computational cost of the proposal is not discussed in the paper. It seems that the dataset needs to be considerably augmented to train the logit predictor (it is trained on the augmented data for all quantile values).

The proposal is not compared to existing alternative calibration strategies.

The writing could be improved.

**Q5 Detailed Comments To The Authors:**

As I mentioned above, the trick used in the paper is clever; yet, the resulting strategy which "turns" probabilistic classifier outputs into calibrated probabilities does not seem to offer strong calibration guarantees. Theorem 3.1 assumes a strong duality property for $Q(\boldsymbol{x},\tau)$, the satisfiability of which is not clear (and could be discussed). Theorem 4.1 shows that the quantile probabilities output by the proposed approach correspond to the actual ones when the base classifier is perfect, which is arguably a very strong assumption. Can you provide any clarification ?

There exist a number of calibration strategies for binary classifiers—see the extensive literature on classifier calibration, or on conformal prediction. The proposed approach should be compared to at least several baselines.

Can you provide additional details on the computational complexity of the proposal ?

The approach seems restricted to binary classifiers, so that the duality property guaranteed by Equation (4) holds. The extension to the multiclass case is not clear: in this case, it is not sure that a similar duality can be attained.

The writing could be improved. The frequent (and most of the time inappropriate) use of italics is harmful to readability.
The fonts are not always appropriate (such as, e.g., the use of "QuantProb" in small caps as an abbreviation for "quantile probabilities"; or "softmax" in sans-serif fonts). There is an inappropriate use of dashes as well: the authors actually use minus signs "-" instead of dashes "--", whereas em dashes "---" should be used as a punctuation mark. Last, but not least,
- the paper uses the writing "equation (n)" in place of "Equation (n)";
- there are several missing words (e.g., 'In binary case", page 3; "We present duality property", page 3; etc);
- there are many typos (e.g., "model suffer greatly", page 2; "Quantile regression techniques [...] offers a wide range of advantages", page 3; "only upto this approximation", page 4; "pre-tained model", page 6; etc).

**Q9 Complying With Reviewing Instructions:**

Yes

---

> ### Author Rebuttal · Authors · 2024-04-02
>
> We thank the reviewer for the comments and hope that the following answers the questions raised.
>
>
> **More Experiments** We now include comparison with Platt Scaling [accuracy](https://anonymous.4open.science/r/quantprob-C207/reports/boxplot_accuracy_tempscaling.png) and [calibration error](https://anonymous.4open.science/r/quantprob-C207/reports/boxplot_calib_error_tempscaling.png). And also on CIFAR100C - [accuracy](https://anonymous.4open.science/r/quantprob-C207/reports/boxplot_accuracy_cifar100.png) and [calibration error](https://anonymous.4open.science/r/quantprob-C207/reports/boxplot_calib_error_cifar100.png). Note that our observations do not change with these additional experiments.
>
> We believe that further comparisons would not change our observations. The main reason is that - Most of the current methods use a valid set to calibrate the probabilities but really do not consider calibration across distortions. However, in this article our main aim is to maintain constant calibration.
>
> **Extension to Multi-Class:** The procedure we adopt to extend the theoretical results empirically is by using a one-vs-rest classification procedure. We agree to some extent that this is a hack (although widely adopted, for instance SVM to Multi-Class-SVM), and it is desirable to have a neat theoretical generalization to higher dimensions (more number of classes).
>
> Quantiles in 1-d are well understood. Section 8.6 in [Koenker, 2005 - Quantile Regression] lists several approaches for computing high dimensional quantiles.  while there is enough result in literature on high dimensional quantile for regression [1],  generalization of quantiles to high dimensions is an open problem and there is no consensus about the right procedure. Hence there isn't one straight forward extension of our proposed approach, and empirically we observe that one-vs-rest works sufficiently well enough.
>
> [1] Tan et.al, Journal of the Royal Statistical Society Series B: Statistical Methodology, Volume 84, Issue 1, February 2022, Pages 205–233
>
> **Computational Cost:** To recall, our procedure assumes a pre-trained classifier $f_{\theta}$ and
> 1. Computing $y_{i,\tau} = I[f_{\theta}(x_i) > 1-\tau]$ - which the naive implementation $O(n\times n_{\tau}$) arguably the most expensive step. However, one can easily adapt this to be more efficient by performing a binary search across $\tau$, giving $O(n \times \log(n\_{\tau}))$.
> 2. Retrain the classifier using $\{(input:(x_i,\tau), output:y_{i,\tau})\} $. While this might look computationally intensive, starting from the pre-trained weights of $f_{\theta}$ allows very fast training. We observe that one can reach convergence in $10-20$ epochs.
>
> **Why did we not consider alternative and SOTA calibration strategies?:** As we state in the article, most of the calibration strategies aim for reducing the calibration error on some hold-out set. This is the standard procedure adopted by scaling and conformal methods.
>
> However, our aim in this article is not to reduce the calibration error at $distortion\ severity=0$, but rather to make it constant across distortions by using only the data at $distortion\ severity=0$. To our knowledge, there is no approach which claims this. Hence, we believe that these comparisons would not be meaningful.
>
> **Stronger theoretical guarantees and Theorem 4.1** We agree that theorem 4.1 assumes that the base classifier is perfect and that it can be considered a strong assumption. The main intention here was to provide sanity checks to the procedure we proposed. However, this theorem does add a lot of value. For instance, the same cannot be said about usual sigmoid probabilities if the error distribution is not symmetric around $0$.
>
> Also, this theorem actually offers the intuition on why quant probs are more robust. Let $X \to X^{+}$ denote a arbitrary distortion. If (again moving away from notation in the article) the labels are given by $\hat{Y}(X) = I[f_{\theta}(X) + \epsilon \geq 0]$, then we know that $(X, \hat{Y}(X))$ and $(X^+,\hat{Y}(X^+))$ both have calibration error $0$ under $f_{\theta}$. This has a "normalizing effect". Now, when these distributions change as $(X, \hat{Y}(X)) \to (X, Y)$ (Y denotes the actual gt-labels), and $(X^{+}, \hat{Y}(X^{+})) \to (X^{+}, Y)$, observe that it is only the labels which change. So, we expect that calibration error differences between these two shifts are approximately the same under small distortions. Hence, on an average we expect the calibration error to remain constant.
>
> While stronger theoretical guarantees are desirable, the fundamental issue is with the no-free-lunch for calibration. Unless we restrict the distortions $X \to X^{+}$ (original to distorted), it does not look possible to provide any guarantees. And as stated before, to our knowledge this issue is not investigated much and there is no consensus on what can be considered "acceptable" restrictions on the possible distortions.

---

### Official Review · Reviewer_77pE · 2024-03-22

**Q2-1 Originality-Novelty:** 3
**Q2-2 Correctness-Technical Quality:** 2
**Q2-5 Clarity Of Writing:** 3

**Q10 Ethical Concerns:**

No ethical concerns

**Q1 Summary And Contributions:**

This paper investigates calibration errors and argues that it is more important to have calibration errors that don't change too much with distortions than small calibration errors. Their argument for this is that a high calibration error can be corrected by tuning the threshold but instability in the calibration errors across distortions can't be fixed. They further propose a method for post-hoc calibration that constructs quantile probabilities for any pre-trained classifier.

**Q2-3 Extent To Which Claims Are Supported By Evidence:**

2: Fair: the main claims are somewhat supported by evidence (but the experimental evaluation may be weak, or does not match entirely with the claims, important baselines may be missing, proofs contain important ideas but lack rigor, algorithmic details are only discussed superficially, references are imprecise, assumptions are not sufficiently motivated or explicated, etc.).

**Q2-4 Reproducibility:**

2: Fair: key resources (e.g. proofs, code, data) are unavailable but key details (e.g. proof sketches, experimental setup) are sufficiently well-described for an expert to confidently reproduce the main results.

**Q3 Main Strengths:**

1. The paper is easy to follow, and its intended claims are clear.
2. The method seems novel and interesting

**Q4 Main Weakness:**

1. I was not able to find sufficient reasoning for why using quantprob to assign probabilities would lead to a more stable calibration error.
2. In the OOD detection experiments Table 1 and Table 2 has no standard deviations. Furthermore, the difference from baselines is very small so I would at least like the results replicated for 5 random seeds.

**Q5 Detailed Comments To The Authors:**

In theorem 4.1 is, I believe, the main result about quantile representations having stable calibration error. It only shows that for a model whose predictions match the labels the quantile representations will match the perfectly calibrated probabilities. I don't see what implication this has for calibration under distortions? Can the authors clarify this further. I think general a more rigorous discussion(starting from a definition of distortion) and the properties of quantile representations under such distortions would greatly enhance the paper.

**Q9 Complying With Reviewing Instructions:**

Yes

---

> ### Author Rebuttal · Authors · 2024-04-02
>
> We thank the reviewer for the valuable comments and hope that the following provides more intuition so as to why QuantProb are more likely to invariant under distortions.
>
> **Why QuantProb are more likely to be invariant under small distortions than standard classifier?** The empirical reasoning we offer in the article is as follows:
>
> 1. While usual classifiers predict the labels with remarkable accuracy, it is known that these classifier are over/underconfident in their probabilities [Guo. et. al., 2017] . Interestingly, under distortions the predictions themselves are relatively decent (for instance in figures 2/3, we see an accuracy > 0.5 for most cases, when random baseline is 0.1).  However, under distortions the calibration error increases (as shown in figures 2/3).
> 2. The main idea behind QuantProb is that, instead of predicting a single class, what we predict is that the probability is greater than $1-\tau$ or not - call this classifier $f_{\theta,\tau}$ (slightly deviating from the notation in article for simplicity)
> 3. If we assume that these classifiers $f_{\theta,\tau}$ can predict  the label well under distortions, then thanks to the framework in the article, taking an average of these predictions give a probability which is robust to distortions.
> 4. This is also the place where we have theorem 4.1. Let $X \to X^{+}$ denote a arbitrary distortion. If (again moving away from notation in the article) the labels are given by $\hat{Y}(X) = I[f_{\theta}(X) + \epsilon \geq 0]$, then we know that $(X, \hat{Y}(X))$ and $(X^+,\hat{Y}(X^+))$ both have calibration error $0$. This has a "normalizing effect". Now, when these distributions change as $(X, \hat{Y}(X)) \to (X, Y)$ (Y denotes the actual gt-labels), and $(X^{+}, \hat{Y}(X^{+})) \to (X^{+}, Y)$, observe that it is only the labels which change. So, we expect that calibration error differences between these two shifts are approximately the same under small distortions. Hence, on an average we expect the calibration error to remain constant.
>
> The key empirical fact is that - Under distortions, while classifiers are good at predicting labels, they are not good at predicting probabilities. From a bird's eye view, what we achieved in this article is to convert the problem of assigning probabilities as a sequence of classification problems using quantiles. This is done by exploiting the duality property and providing an efficient framework for the same.
>
> For a more concrete intuition consider an hypothetical extension to figure 1 in the article. In [img1](https://anonymous.4open.science/r/quantprob-C207/img/Figure_scatter.png) we distort the dataset slightly by adding translations and compute the calibration errors in [img2](https://anonymous.4open.science/r/quantprob-C207/img/Figure_calib.png). The same behaviour as observed in CIFAR10C is repeated here as well.
>
> Do note that this behaviour is only for a combination of meaningful classifiers and meaningful distortions. In general, because of no-free-lunch in calibration, we cannot provide rigorous assurances. Unless, we know a reasonable assumptions to make on the possible set of distortions.
>
> **OOD Detection:** We agree with the reviewer that the results are very close to MLS and MSP. In fact, we claim that our method  is very close to MLS except that we measure in terms of quantiles instead of actual logits, leading to small but negligible deviations. And moreover MLS (to our knowledge) is the current SOTA. We shall include the stdev. as well in the final version.
>
> **More Experiments** We now include comparison with Platt Scaling [accuracy](https://anonymous.4open.science/r/quantprob-C207/reports/boxplot_accuracy_tempscaling.png) and [calibration error](https://anonymous.4open.science/r/quantprob-C207/reports/boxplot_calib_error_tempscaling.png). And also on CIFAR100C - [accuracy](https://anonymous.4open.science/r/quantprob-C207/reports/boxplot_accuracy_cifar100.png) and [calibration error](https://anonymous.4open.science/r/quantprob-C207/reports/boxplot_calib_error_cifar100.png). Note that our observations do not change with these additional experiments.

---

### Official Review · Reviewer_S4jC · 2024-03-22

**Q2-1 Originality-Novelty:** 3
**Q2-2 Correctness-Technical Quality:** 3
**Q2-5 Clarity Of Writing:** 4

**Q1 Summary And Contributions:**

The authors aimed to improve the quantile regression problem by introducing QuantProb, motivated by the duality observation of the SQR formulation. Even though the method implicitly introduce a dependency to the underlying (uncalibrated) classifier, it is relatively straightforward to actually implement.  Experiment on CIFAR-10-C, i.e., perturbed CIFAR-10, shows similar calibration errors across varying level of perturbation, showing the effectiveness of QuantProb.

**Q2-3 Extent To Which Claims Are Supported By Evidence:**

2: Fair: the main claims are somewhat supported by evidence (but the experimental evaluation may be weak, or does not match entirely with the claims, important baselines may be missing, proofs contain important ideas but lack rigor, algorithmic details are only discussed superficially, references are imprecise, assumptions are not sufficiently motivated or explicated, etc.).

**Q2-4 Reproducibility:**

4: Excellent: key resources (e.g. proofs, code, data) are available and key details (e.g. proof sketches, experimental setup) are comprehensively described for competent researchers to confidently and easily reproduce the main results.

**Q3 Main Strengths:**

- The paper is exceptionally well-written, easily the top 10% of usual UAI papers. The formulation was straightforward to follow, as the authors provided numerous helpful remarks and discussions to extend the understanding of the reader.
- The method motivation on dualities were quite interesting, potentially further sparking future works.
- Raising the issue of having similar calibration quality per distortion is also interesting, which should be considered more.

**Q4 Main Weakness:**

Experiments can be improved:
The authors only considered the maximum softmax probability (MSP) as the baseline, where it is widely known that they are quite bad in calibration. Even though the method is motivated by/upgraded existing methods, i.e., pinball loss or SQR, they are not used as the baseline, among with other confidence calibration baselines.

Clarify the limitations:
- In real-world settings, it is sensible to expect two training runs for applying this method, which requires x2 computation, unlike SQR.
- The modified labels $y_{i,\tau}^+$ would be inherently dependent on the classifier. Especially, the method seems to use the training set, and $\tau$ will likely be quite high (extremely concentrated around 1.0) due to overconfidence issues of neural nets. However, it is likely that the distribution of the probability values of seen and unseen samples would be different. I couldn't see how the method was avoiding this problem.
- Even though the authors mention the No Free Lunch, I do not see how QuantProb directly avoids this problem in varying distortion levels. Can the authors clarify how exactly the method shows similar ECEs?

**Q5 Detailed Comments To The Authors:**

Detailed suggestions for the experiments:
- I understand that especially pinball loss requires training per tau, but we could have feasible settings, such as number of bins to be 5. Note that QuantProb also requires two training runs.
- Simple methods outside QR should be also considered, such as temperature scaling, which would've been easy enough baseline to test, given that the authors already provide MSP as the baseline.
- ResNet and Densenet do not show big differences in trends, so authors could add other baselines while pushing one of those to the Appendix.
- Figure 2 and 3's legend order and color is different. I was quite confused when I first encountered these two figures.
- In Figure 4, only QuantProb is considered. This figure should also include MSP. I think MSP would benefit greatly for Platt scaling.
- It is more common to also draw accuracy per bin (similar to Guo et al) because (i) it would be easy to maximize ECE only, as one can fill up one specific bin with all the samples per valid set accuracy (even though this is not the case for this paper, as they also showed accuracy), and (ii) can observe the behaviors per bin.

**Q9 Complying With Reviewing Instructions:**

Yes

---

> ### Author Rebuttal · Authors · 2024-04-02
>
> We thank the reviewer for the valuable comments.
>
> **Computation of QuantProb:** We agree that in principle one needs 2 training runs for QuantProb. However, note that - After the first training, one can start the second train from the parameters of first training, essentially fine tuning for probabilities only. This would lead to quick convergence and hence no significant overhead.
>
> **Why not SQR directly?** Apart from training for each $\tau$ as the reviewer points out, SQR loss have several known issues - (i) SQR loss is difficult to optimize in classification settings (ii) In general, especially in current workflows, it is common that one starts with networks trained and finetuned using different techniques. Our approach preserves the information from the pretrained model, while training with SQR from ground-truth labels could potentially remove existing information.
>
> **Comparison with Platt Scaling:** Note that our aim is to maintain the calibration error across distortions.  Techniques such as Platt-Scaling, while reducing the calibration error on the original distribution, do not really effect the behaviour on distorted distributions. This is because, these techniques correct the probabilities on a "valid set" and overfit on this set to ensure low calibration errors.
>
> We observe that Platt-Scaling, while improving the robustness to distortions slightly, still do not maintain the constant calibration across distortions.The figures are at [img1](https://anonymous.4open.science/r/quantprob-C207/reports/boxplot_accuracy_tempscaling.png) which shows the accuracy, and [img2](https://anonymous.4open.science/r/quantprob-C207/reports/boxplot_calib_error_tempscaling.png).
>
> **More experiments** We include the experiment on CIFAR100C [link](https://anonymous.4open.science/r/quantprob-C207/reports/boxplot_calib_error_cifar100.png) and [link](https://anonymous.4open.science/r/quantprob-C207/reports/boxplot_accuracy_cifar100.png) and verify that our observations hold in this case as well.
>
> **Why QuantProb are more likely to be invariant under small distortions than standard classifier?** The empirical reasoning we offer in the article is as follows:
>
> 1. While usual classifiers predict the labels with remarkable accuracy, it is known that these classifier are over/underconfident in their probabilities [Guo. et. al., 2017] . Interestingly, under distortions the predictions themselves are relatively decent (for instance in figures 2/3, we see an accuracy > 0.5 for most cases, when random baseline is 0.1).  However, under distortions the calibration error increases (as shown in figures 2/3).
> 2. The main idea behind QuantProb is that, instead of predicting a single class, what we predict is that the probability is greater than $1-\tau$ or not - call this classifier $f_{\theta,\tau}$ (slightly deviating from the notation in article for simplicity)
> 3. If we assume that these classifiers $f_{\theta,\tau}$ can predict  the label well under distortions, then thanks to the framework in the article, taking an average of these predictions give a probability which is robust to distortions.
> 4. This is also the place where we have theorem 4.1. Let $X \to X^{+}$ denote a arbitrary distortion. If (again moving away from notation in the article) the labels are given by $\hat{Y}(X) = I[f_{\theta}(X) + \epsilon \geq 0]$, then we know that $(X, \hat{Y}(X))$ and $(X^+,\hat{Y}(X^+))$ both have calibration error $0$. This has a "normalizing effect". Now, when these distributions change as $(X, \hat{Y}(X)) \to (X, Y)$ (Y denotes the actual gt-labels), and $(X^{+}, \hat{Y}(X^{+})) \to (X^{+}, Y)$, observe that it is only the labels which change. So, we expect that calibration error differences between these two shifts are approximately the same under small distortions. Hence, on an average we expect the calibration error to remain constant.
>
> The key empirical fact is that - Under distortions, while classifiers are good at predicting labels, they are not good at predicting probabilities. From a bird's eye view, what we achieved in this article is to convert the problem of assigning probabilities as a sequence of classification problems using quantiles. This is done by exploiting the duality property and providing an efficient framework for the same.
>
> For a more concrete intuition consider an hypothetical extension to figure 1 in the article. In [img1](https://anonymous.4open.science/r/quantprob-C207/img/Figure_scatter.png) we distort the dataset slightly by adding translations and compute the calibration errors in [img2](https://anonymous.4open.science/r/quantprob-C207/img/Figure_calib.png). The same behaviour as observed in CIFAR10C is repeated here as well.
>
> Do note that this behaviour is only for a combination of meaningful classifiers and meaningful distortions. In general, because of no-free-lunch in calibration, we cannot provide rigorous assurances. Unless, we know a reasonable assumptions to make on the possible set of distortions.

---

### Official Review · Reviewer_pHb1 · 2024-03-22

**Q2-1 Originality-Novelty:** 2
**Q2-2 Correctness-Technical Quality:** 3
**Q2-5 Clarity Of Writing:** 3

**Q1 Summary And Contributions:**

The paper is concerned with learning the quantile function for a classification problem. In particular the authors propose a method to learn the quantiles from a pre-trained classifier. Their approach is based on the duality between "quantile representation" and "probability representation" stemming from the use of the pinball loss to learn the quantile function.

**Q2-3 Extent To Which Claims Are Supported By Evidence:**

3: Good: the main claims are supported by convincing evidence (in the form of adequate experimental evaluation, proofs, (pseudo-)code, references, assumptions).

**Q2-4 Reproducibility:**

3: Good: key resources (e.g. proofs, code, data) are available and key details (e.g. proofs, experimental setup) are sufficiently well-described for competent researchers to confidently reproduce the main results.

**Q3 Main Strengths:**

The method proposed by the authors seems sound and clearly explained, and is well supported by experimental data.

**Q4 Main Weakness:**

- The proposed method relies on refitting a model for each quantile $\tau$, and it seems that a large number of quantiles is required to properly approximate the integral in (13). Hence, I would be curious to know the computational cost of QuantProb.
- While the performance of the method depends on the original confidence scores $f_{\theta}$, I think the paper lacks a description / experiments on the impact of the quality of these confidence scores on the performance of QuantProb,

**Q5 Detailed Comments To The Authors:**

- While I understand that Figure 1 is just an illustration, I fail to see how it shows that QuantProb improves uncertainty quantification. Indeed, in this example the data is essentially noiseless so the confidence should be either 0 or 1. In other words, it seems that using your method will artificially (and wrongly) reduce the confidence score on these data points, is my intuition correct ?
- The related work section  is missing some reference on quantile regression e.g. arxiv:2106.05515, where the authors discuss the under-coverage bias of quantile regression in high-dimension.

Question:
- Why is it necessary to assume that the hypothesis class is able to fit the training data (Remark in section 3.1) ?
- There is no comparison with temperature scaling in the experimental part, is there any reason for this ?
- The performance of QuantProb is surely dependent on the original confidence scores $f_{\theta}$ of the pre-trained model. In your experiments you directly use the pretrained NN, what would be the performance of your method if you first recalibrate these scores (e.g with temperature scaling) ?

**Q9 Complying With Reviewing Instructions:**

Yes

---

> ### Author Rebuttal · Authors · 2024-04-02
>
> We thank the reviewer for the valuable comments and the references.
>
> **Why QuantProb are more likely to be invariant under small distortions than standard classifier?** The empirical reasoning we offer in the article is as follows:
>
> 1. While usual classifiers predict the labels with remarkable accuracy, it is known that these classifier are over/underconfident in their probabilities [Guo. et. al., 2017] . Interestingly, under distortions the predictions themselves are relatively decent (for instance in figures 2/3, we see an accuracy > 0.5 for most cases, when random baseline is 0.1).  However, under distortions the calibration error increases (as shown in figures 2/3).
> 2. The main idea behind QuantProb is that, instead of predicting a single class, what we predict is that the probability is greater than $1-\tau$ or not - call this classifier $f_{\theta,\tau}$ (slightly deviating from the notation in article for simplicity)
> 3. If we assume that these classifiers $f_{\theta,\tau}$ can predict  the label well under distortions, then thanks to the framework in the article, taking an average of these predictions give a probability which is robust to distortions.
> 4. This is also the place where we have theorem 4.1. Let $X \to X^{+}$ denote a arbitrary distortion. If (again moving away from notation in the article) the labels are given by $\hat{Y}(X) = I[f_{\theta}(X) + \epsilon \geq 0]$, then we know that $(X, \hat{Y}(X))$ and $(X^+,\hat{Y}(X^+))$ both have calibration error $0$. This has a "normalizing effect". Now, when these distributions change as $(X, \hat{Y}(X)) \to (X, Y)$ (Y denotes the actual gt-labels), and $(X^{+}, \hat{Y}(X^{+})) \to (X^{+}, Y)$, observe that it is only the labels which change. So, we expect that calibration error differences between these two shifts are approximately the same under small distortions. Hence, on an average we expect the calibration error to remain constant.
>
> The key empirical fact is that - Under distortions, while classifiers are good at predicting labels, they are not good at predicting probabilities. From a bird's eye view, what we achieved in this article is to convert the problem of assigning probabilities as a sequence of classification problems using quantiles. This is done by exploiting the duality property and providing an efficient framework for the same.
>
> For a more concrete intuition consider an hypothetical extension to figure 1 in the article. In [link](https://anonymous.4open.science/r/quantprob-C207/img/Figure_scatter.png) we distort the dataset slightly by adding translations and compute the calibration errors in [link](https://anonymous.4open.science/r/quantprob-C207/img/Figure_calib.png). The same behaviour as observed in CIFAR10C is repeated here as well.
>
> Do note that this behaviour is only for a combination of meaningful classifiers and meaningful distortions. In general, because of no-free-lunch in calibration, we cannot provide rigorous assurances. Unless, we know a reasonable assumptions to make on the possible set of distortions.
>
> **Comparison with Platt Scaling:** Note that our aim is to maintain the calibration error across distortions. We still maintain that existing techniques such as Platt-Scaling, while reducing the calibration error on the original distribution, do not really effect the behaviour on distorted distributions. This is because, these techniques correct the probabilities on a "valid set" and overfit on this set to ensure low calibration errors.
>
> We observe that Platt-Scaling, while improving the robustness to distortions slightly, still do not maintain the constant calibration across distortions.The figures are at [link](https://anonymous.4open.science/r/quantprob-C207/reports/boxplot_accuracy_tempscaling.png) which shows the accuracy, and [link](https://anonymous.4open.science/r/quantprob-C207/reports/boxplot_calib_error_tempscaling.png).
>
> **In your experiments you directly use the pretrained NN, what would be the performance of your method if you first recalibrate these scores (e.g with temperature scaling)?** Since most calibration techniques preserve the order of the points, and practically we consider the logits directly (eq(12) in the article), we do not expect that re-calibrating the scores will result in any major differences.
>
> **Does our method artificially (and wrongly) reduce the confidence score on data points?** Yes, the intuition that our method reduces the confidence score is correct. However, we would like to argue that this is not wrong, just different. In the article we show that QuantProb is more robust to distortions than usual probabilities and also has properties as indicated by theorem 4.1.
>
> **Why is it necessary to assume that the hypothesis class is able to fit the training data (Remark in section 3.1) ?** We assume this in the intuitive explanations we give in the rest of the article. Allows us to explain the key ideas better.

---

### Meta-Review · Area_Chair_oc2t · 2024-04-23

Coming soon.